# Predictable Reinforcement Learning Dynamics through Entropy Rate Minimization

**Daniel Jarne Ornia**$^{*\dagger}$ 
*University of Oxford*
*daniel.jarneornia@ox.cs.ac.uk*

**Giannis Delimpaltadakis**$^{*}$
*Eindhoven University of Technology*

**Jens Kober**
*Delft University of Technology*

**Javier Alonso-Mora**
*Delft University of Technology*

**Reviewed on OpenReview:** *https://openreview.net/forum?id=DDUsc1lD27*

## Abstract

In Reinforcement Learning (RL), agents have no incentive to exhibit predictable trajectories, and are often pushed (through e.g. policy entropy regularisation) to randomise their actions in favor of exploration. This lack of predictability awareness often makes it challenging for other agents and humans to predict an agent's trajectories, possibly triggering unsafe scenarios (e.g. in human-robot interaction). We propose a novel method to induce predictable trajectories in RL agents, termed *Predictability-Aware RL* (PARL), employing the agent's trajectory *entropy rate* to quantify predictability. Our method maximizes a linear combination of a standard discounted reward and the negative entropy rate, thus trading off optimality with predictability. We show how the entropy rate can be formally cast as an average reward, how entropy-rate value functions can be estimated from a learned model and incorporate this in policy-gradient algorithms, and demonstrate how this approach produces predictable (near-optimal) policies in tasks inspired by human-robot use-cases.

## 1 Introduction

As Reinforcement Learning (RL) (Sutton & Barto, 2018) agents are deployed to interact with humans, it becomes crucial to ensure that their behaviours[1] are predictable. A robot trained under general RL algorithms operating in a human environment has no incentive to follow trajectories that are easy to predict. This makes it challenging for other robots or humans to forecast the robot's behaviour, affecting coordination and interactions, and possibly triggering unsafe scenarios. RL algorithms are oblivious to the predictability of behaviours they induce in agents: one aims to maximize an expected reward, regardless of how unpredictable trajectories taken by the agents may be. In fact, many algorithms propose some form of regularisation in

---

$^{*}$Equal contribution.
$^{\dagger}$Work done while at Delft University of Technology.
[1]We use agent behaviour to refer to, throughout this work, the state trajectories agents exhibit (and observers may perceive). We consider 'agent behaviour' or 'agent trajectory' interchangeably, but note that we mainly focus on state predictability. We make the case that an agent is *predictable* if their next state (for a fixed policy) is easy to predict given their past state(s) for standard inference algorithms.

action complexity (Schulman et al., 2017; Han & Sung, 2021) or value functions (Pitis et al., 2020; Zhao et al., 2019; Kim et al., 2023) for better exploration, inducing higher aleatoric uncertainty in agents' trajectories.

We quantify predictability of an RL agent's trajectories by employing the notion of *entropy rate*: the infinite-horizon time-average entropy of the agent's trajectories, which measures the complexity of the trajectory distributions induced in RL agents. Higher entropy rate implies more complex and less predictable trajectories, and vice-versa. Similar information-theoretic metrics have been widely used to quantify (un)predictability of stochastic processes Shannon (1948); Savas et al. (2022); Biondi et al. (2014); Duan et al. (2019); Stefansson & Johansson (2021).

**Motivation**   In general, RL agents are oblivious to the information theoretic loads they generate with their behaviour. In a world where agents do not exist in a vacuum (even if we train RL agents in single-agent settings, these agents will rarely be deployed in isolation), one could argue there is an advantage to inducing a *complexity awareness* in RL agents; *If an agent can solve a task generating lower information rates, it should do so.* Lower information rates correspond (intuitively and formally, through forms of entropy) with lower uncertainty. However, we do not argue that this is a necessary feature in all agents (or even always desireable). We simply argue that it is an interesting feature to consider for general RL agents that can benefit the deployment of RL agents, propose a formal approach to target this, and evaluate how this impacts such agents in different settings.

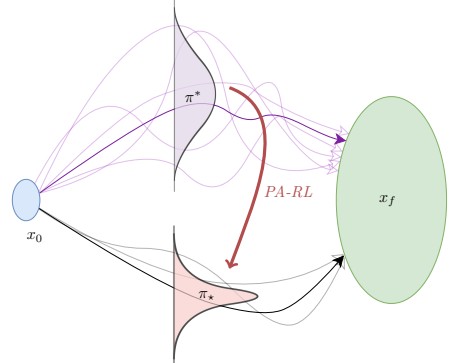

Figure 1: Qualitative representation of PARL. Trajectories are represented symbolically as connecting an initial with a final set of states. PARL shifts the policies towards smaller trajectory entropy.

**Entropy and Predictability**   The established notion for predictability in information theory is entropy: a lower-entropy stochastic process is easier to predict by standard inference algorithms. For example, assuming the trajectory distribution is Gaussian, lower entropy is equivalent to lower variance. Then it is clear that (inside the family of normal distributions) inference algorithms would be able to predict more accurately a lower variance distribution. The same applies to other distribution families, including discrete distributions. In the limit, minimum entropy implies that the agent follows the same deterministic trajectory over and over, which means it can be predicted with no prediction error from inference on past data (since it is deterministic).

**Contributions**   We propose a novel approach to model-based RL that induces more predictable behaviour in RL agents, termed *Predictability-Aware RL* (PARL). We maximize the linear combination of a standard discounted cumulative reward and the negative entropy rate, thus trading-off optimality with predictability. Towards this, we cast entropy-rate minimization as an expected average reward minimization problem, with a policy-dependent reward function, called *local entropy*. To circumvent local entropy's policy-dependency and enable the use of on- and off-policy RL algorithms, we introduce a state-action-dependent surrogate entropy, and show that deterministic policies minimizing the average surrogate entropy exist and *also minimize the entropy rate*. Further, we show how, employing a learned model and the surrogate reward, we can estimate entropy-rate value functions, and incorporate this in policy-gradient schemes. Finally, we showcase how PARL produces much more predictable agents while achieving near-optimal rewards in several robotics and autonomous driving tasks[2].

## 1.1   Related work

The idea of introducing some form of entropy objectives in policy gradient algorithms has been extensively explored Williams & Peng (1991); Fox et al. (2015); Peters et al. (2010); Zimin & Neu (2013); Neu et al.

---

[2]See the project repository https://github.com/tud-amr/parl for details.

(2017); Tiapkin et al. (2023). In most instances, these regularization terms are designed to either help policy randomization and exploration Mutti et al. (2021; 2022), or to stabilize RL algorithms. However, these works focus on policy (state-action) entropy maximization, and do not focus on trajectory entropy and how it affects predictability of RL agents.

Instead, Guo et al. (2023b;a); Biondi et al. (2014); George et al. (2018); Duan et al. (2019); Stefansson & Johansson (2021); Savas et al. (2020; 2022) consider entropy (rate) maximization in (PO)MDPs, to yield unpredictable behaviours. However, these works require full knowledge of the model, and entropy (rate) maximization is cast as a non-linear program. Instead, in our work, the model is not known, and we resort to learning entropy-rate value functions.

Recent work Lu et al. (2020); Eysenbach et al. (2021); Park & Levine (2023) has tackled robustness and generalization via introducing Information-Theoretic penalty terms in the reward function. In particular, Eysenbach et al. 2021 makes the explicit connection from such information-theoretic penalties to the emergence of predictable behaviour in RL agents, and uses mutual information penalties to restrict the bits of information that the agents are allowed to use, resulting in simpler, less complex policies. We address directly this predictability problem by the minimisation of entropy rates in RL agents' stochastic dynamics. This allows us to provide theoretical results regarding existence and convergence of optimal (predictable) policies towards *minimum entropy-rate* agents, and make our scheme generalizeable to *any RL algorithm* (on and off policy). In this line, Berseth et al. (2021) propose *Surprise Minimizing Reinforcement Learning*, where an estimate of the state stationary distribution of visited states is kept, and the agents are penalised for visiting states with low probabilities in this stationary distribution. While having a similar flavour, our work addresses *state trajectory* entropy, contrary to stationary distribution entropy, since we aim to address the predictability of the agent's data generating process (and not simply avoid unknown and changing environment regions)[3]. Finally, our work is tangentially related to alignment and interpretability in RL Shah et al. (2019); Carroll et al. (2019), where human-agent interaction requires exhibiting human-interpretable behaviour. Further, work on legibility of robot motion Dragan et al. (2013); Liu et al. (2023); Busch et al. (2017) shares our motivation; to make robotic systems behaviour more legible by humans.

## 2 Background

We, first, introduce preliminary concepts employed throughout this work. For more detail on Markov chains and decision processes, the reader is referred to Puterman (2014). Given a set $\mathcal{A}$, $\Delta(\mathcal{A})$ denotes the probability simplex over $\mathcal{A}$, and $\mathcal{A}^k$ denotes the $k$-times Cartesian product $\mathcal{A} \times \mathcal{A} \times \cdots \times \mathcal{A}$. If $\mathcal{A}$ is finite, $|\mathcal{A}|$ denotes its cardinality. Given two probability distributions $\mu, \nu$, we use $D_{TV}(\mu\|\nu)$ as the total variation distance between two distributions. We use $\text{supp}(\mu)$ to denote the support of $\mu$. Given two vectors $\xi, \eta$, we write $\xi \succeq \eta$, if each $i$-th entry of $\xi$ is bigger or equal to the $i$-th entry of $\eta$.

### 2.1 Markov processes and Rewards

A Markov Chain (MC) is a tuple $\mathcal{C} = (\mathcal{X}, P, \mu_0)$ where $\mathcal{X}$ is a finite set of states, $P : \mathcal{X} \times \mathcal{X} \to [0,1]$ is a transition probability measure and $\mu_0 \in \Delta(\mathcal{X})$ is a probability distribution over initial states. Specifically, $P(x,y)$ is the probability of transitioning from state $x$ to state $y$. $P^t(x,y)$ is the probability of landing in $y$ after $t$ time-steps, starting from $x$. The limit transition function is $P^* := \lim_{t\to\infty} P^t$. We use uppercase $X_t$ to refer to the random variable that is the state of the random process governed by the MC at time $t$, and lower case to indicate specific states, *e.g.* $x \in \mathcal{X}$. Similarly, a trajectory or a path is a sequence of states $\mathbf{x}_k = \{x_0, x_1, ..., x_k\}$, where $x_i \in \mathcal{X}$, and $\mathcal{P}_k$ denotes the set of all $(k+1)$-length paths. We also denote $X_{t:k} = \{X_t, X_{t+1}, ..., X_k\}$. Further, we define $p : \mathcal{P}_\infty \to [0,1]$ as a probability measure over the Borel $\sigma$-algebra $\mathcal{B}(X_{0:\infty})$ of infinite-length paths of a MC[4] conditioned to initial distribution $\mu_0$. For example, $p(X_0 = x) \equiv \mu_0(x)$ is the probability of the initial state being $x$; $p(X_{0:3} = \mathbf{x}_3)$ is the probability that the MC's state follows the path $\mathbf{x}_3$. A Markov Decision Process (MDP) is a tuple $\mathcal{M} = (\mathcal{X}, \mathcal{U}, P, R, \mu_0)$ where $\mathcal{X}$ is a finite set of states, $\mathcal{U}$ is a finite set of actions, $P : \mathcal{X} \times \mathcal{U} \times \mathcal{X} \to [0,1]$ is a probability measure

---

[3]A simple counterexample can show that an agent can have (almost) *zero trajectory entropy* and have high stationary distribution entropy; think of two connected states $A \leftrightarrow B$ where $A$ has one self loop with near to zero probability.

[4]This measure is well defined by the Ionescu-Tulcea Theorem, see *e.g.* Dudley (2018).

of the transitions between states given an action, $R : \mathcal{X} \times \mathcal{U} \times \mathcal{X} \to \mathbb{R}$ is a bounded reward function and $\mu_0 \in \Delta(\mathcal{X})$ is the probability distribution of initial states. A stationary Markov policy is a stochastic kernel $\pi : \mathcal{X} \to \Delta(\mathcal{U})$. With abuse of notation, we use $\pi(u \mid x)$ as the probability of taking action $u$ at state $x$, under policy $\pi$. Let $\Pi$ be the set of all stationary Markov policies, and $\Pi^D \subseteq \Pi$ the set of deterministic policies. The composition of an MDP $\mathcal{M}$ and a policy $\pi \in \Pi$ generates a MC with transition probabilities $P_\pi(x, y) := \sum_{u \in \mathcal{U}} \pi(u \mid x) P(x, u, y)$. If said MC admits a unique stationary distribution, we denote it by $\mu^\pi$, where $\mu^\pi : \mathcal{X} \to [0, 1]$. We, also, use the shorthand $R_t^\pi \equiv \mathbb{E}_{u \sim \pi(x)}[R(X_t, u, X_{t+1})]$. We will assume any fixed policy $\pi$ in MDP $\mathcal{M}$ induces an aperiodic and irreducible MC.

**Discounted Reward MDPs** In discounted cumulative reward problems the goal is to find a policy $\pi'$ that maximizes the discounted sum of rewards for discount factor $\gamma \in [0, 1)$: i.e. $\pi' \in \arg\max_{\pi \in \Pi} \mathbb{E}[\sum_{t=0}^\infty \gamma^t R_t^\pi \mid X_0 = x]$, for all $x \in \mathcal{X}$. In this case Puterman (2014) given a policy $\pi$, the *value function* under $\pi$, $V^\pi : \mathcal{X} \mapsto \mathbb{R}$, is $V^\pi(x) := \mathbb{E}[\sum_{t=0}^\infty \gamma^t R_t^\pi \mid X_0 = x]$. The *action-value function* (or $Q$-function) under $\pi$ is given by $Q^\pi(x, u) := \sum_{y \in \mathcal{X}} P(x, u, y)(R(x, u, y) + \gamma V^\pi(y))$.

**Average Reward MDPs** In average reward maximization problems, we aim at maximizing the *reward rate* (or gain) $g^\pi(x)$, defined as, together with the *bias*[5]

$$g^\pi(x) := \mathbb{E}\left[\lim_{T \to \infty} \frac{1}{T} \sum_{t=0}^{T} R_t^\pi \mid X_0 = x\right], \quad b^\pi(x) := \mathbb{E}\left[\lim_{T \to \infty} \sum_{t=0}^{T} (R_t^\pi - g(X_t)) \mid X_0 = x\right]. \quad (1)$$

Note that the bias is the expected difference between the stationary rate and the rewards obtained by initialising the system at a given state. For $\pi \in \Pi$, the average-reward (action) value-functions[6] $V_{\mathrm{avg}}^\pi : \mathcal{X} \to \mathbb{R}$ is defined by Abounadi et al. (2001) $V_{\mathrm{avg}}^\pi(x) := \mathbb{E}_{u \sim \pi, y \sim P(x, u, \cdot)}[R(x, u, y) - g^\pi + V_{\mathrm{avg}}^\pi(y)]$ and $Q_{\mathrm{avg}}^\pi(x, u) := \mathbb{E}_{y \sim P(x, u, \cdot)}\left[R(x, u, y) - g^\pi + V_{\mathrm{avg}}^\pi(y)\right]$. The optimal (action) value functions (which exists for ergodic MDPs) satisfy $V_{\mathrm{avg}}^*(x) = \max_{u \in \mathcal{U}} \mathbb{E}_{y \sim P(x, u, \cdot)}\left[R(x, u, y) - g^* + V_{\mathrm{avg}}^*(y)\right]$ and $Q_{\mathrm{avg}}^*(x, u) = \mathbb{E}_{y \sim P(x, u, \cdot)}\left[R(x, u, y) - g^* + V_{\mathrm{avg}}^*(y)\right]$ where $g^*$ is the optimal reward rate.

## 2.2 Shannon Entropy and MDPs

For a discrete random variable $A$ with finite support $\mathcal{A}$, Shannon entropy Shannon (1948) is a measure of uncertainty induced by its distribution, and it is defined as $h(A) := -\sum_{a \in \mathcal{A}} \Pr(A = a) \log(\Pr(A = a))$. Shannon entropy measures the amount of information encoded in a random variable: a uniform distribution maximizes entropy (minimal information), and a Dirac distribution minimizes it (maximal information). Recall $p : \mathcal{P}_\infty \to [0, 1]$ is a probability measure over the Borel $\sigma$-algebra of infinite-length paths of a MC. Let us denote the dependency of $p$ on a fixed policy $\pi$ as $p^\pi$ (a fixed policy in a MDP induces a MC For an MDP under policy $\pi$, recall Section 2.1). Then we define the conditional entropy (Cover, 1999; Biondi et al., 2014) of $X_{T+1}$ given $X_{0:T}$ as:

$$h^\pi(X_{T+1} \mid X_{0:T}) := -\sum_{y \in \mathcal{X}, \mathbf{x}_T \in \mathcal{X}^T} p^\pi(X_{T+1} = y, X_{0:T} = \mathbf{x}_T) \log(p^\pi(X_{T+1} = y \mid X_{0:T} = \mathbf{x}_T)),$$

and the joint entropy[7] of the path $X_{0:T}$ is $h^\pi(X_{0:T}) := h(X_0) + \sum_{t=1}^{T} h^\pi(X_t \mid X_{0:t-1})$.

**Definition 1** (Entropy Rate Shannon (1948))**.** *Whenever the limit exists, the entropy rate of an MDP $\mathcal{M}$ under policy $\pi$ is defined by $\bar{h}^\pi := \lim_{T \to \infty} \frac{1}{T} h^\pi(X_{0:T})$.*

The entropy rate represents the rate of *diversity* in the information generated by the induced MC's paths. Smaller entropy rates imply more predictable trajectories of the induced MC.

---

[5]The bias can also be written in vector form as $b = (I - P + P^*)^{-1}(I - P^*)R^\pi$ where $R^\pi \in \mathbb{R}^{|\mathcal{X}|}$ is the vector of state rewards, $R_x^\pi = \mathbb{E}_{u \sim \pi}[R(x, u, y)]$. See Chapter 8 in Puterman (2014).

[6]Observe that, for ergodic MDPs, $b^\pi(x) = V_{\mathrm{avg}}^\pi \pi(x)$.

[7]The second equality is obtained by applying the general product rule to the joint probabilities of $\mathcal{Y}_T$.

## 3 Entropy Rates: Estimation and Learning

**Problem Statement**   The problem considered in this work is the following. Consider an unknown MDP $\mathcal{M} = (\mathcal{X}, \mathcal{U}, P, R, \mu_0)$. Further, assume that we can sample transitions $\big(x, u, y, R(x, u, y)\big)$ applying any action $u \in \mathcal{U}$ and letting $\mathcal{M}$ evolve according to $P(x, u, \cdot)$. We want

$$\pi_\star \in \arg\max_{\pi \in \Pi} \mathbb{E}[\sum_{t=0}^{\infty} \gamma^t R_t^\pi] - k\bar{h}^\pi, \tag{2}$$

where $k > 0$ is a tuneable parameter[8]. In words, we are looking for policies that maximize a tuneable weighted linear combination of the negative entropy $-\bar{h}^\pi$ and a standard expected discounted cumulative reward. As such, *we establish a trade-off, which is tuned via the parameter k, between entropy rate minimization (i.e. predictability) and optimality w.r.t. the cumulative reward.*

**Proposed approach**   We first show how the entropy rate $\bar{h}^\pi$ can be treated as an average reward criterion, with the so-called *local entropy* $l^\pi$ as its corresponding local reward. Then, because $l^\pi$ is policy-dependent, we introduce a surrogate reward, that solely depends on states and actions and can be learned in-the-loop. We show that deterministic policies minimizing the expected average surrogate reward exist and also minimize the actual entropy rate. Moreover, we prove that, given a learned model of the MDP, we are able to (locally optimally) approximate the value function associated to the entropy rate, via learning the surrogate's value functions. Based on these results, we propose a (model-based[9]) RL algorithm with its maximization objective being the combination of the cumulative reward and an average reward involving the surrogate local entropy.

**Estimating Entropy Rates**   Towards writing the entropy rate $\bar{h}^\pi$ as an expected average reward, let us define the *local entropy*, under policy $\pi$, for state $x \in \mathcal{X}$ as

$$l^\pi(x) := - \sum_{y \in X} P_\pi(x, y) \log P_\pi(x, y). \tag{3}$$

Now, making use of the Markov property, the entropy rate for an MDP reduces to

$$\bar{h}^\pi = \lim_{T \to \infty} \frac{1}{T} \left( h(X_0) + \sum_{t=1}^{T} h^\pi(X_t \mid X_{0:t-1}) \right) = \lim_{T \to \infty} \frac{1}{T} \left( h(X_0) + \sum_{t=1}^{T} h^\pi(X_t \mid X_{t-1}) \right). \tag{4}$$

Observe now, from the definition of $p^\pi$ and again the Markov property,

$$h^\pi(X_t \mid X_{t-1}) = - \sum_{y \in \mathcal{X}, x \in \mathcal{X}} p^\pi(X_t = y, X_{t-1} = x) \log p^\pi(X_t = y \mid X_{t-1} = x)$$

$$= - \sum_{y \in \mathcal{X}, x \in \mathcal{X}} p^\pi(X_{t-1} = x) P_\pi(x, y) \log p^\pi(X_t = y \mid X_{t-1} = x) = \mathbb{E} \left[ - \sum_{y \in \mathcal{X}} P_\pi(x, y) \log P_\pi(x, y) \right].$$

Therefore, substituting in equation 4,

$$\bar{h}^\pi = \lim_{T \to \infty} \frac{1}{T} \sum_{t=0}^{T} \mathbb{E}[l^\pi(X_t)] = \mathbb{E}\big[ \lim_{T \to \infty} \frac{1}{T} \sum_{t=0}^{T} l^\pi(X_t) \big]. \tag{5}$$

We can treat the local entropy $l^\pi$ under policy $\pi$ as a *policy-dependent reward (or cost) function*, since $l^\pi$ is stationary, history independent and bounded. Thus, $\bar{h}^\pi$ is treated as an expected average reward, with reward function $l^\pi$.

---

[8]We choose to cast the problem as a maximization of a linear combination of objectives to allow agents to find efficient trade-offs. This problem could be solved through *e.g.* multi-objective optimization methods Skalse et al. (2022); Hayes et al. (2022). We leave this as an application-dependent choice.

[9]We use the term *model-based*, since we require learning a (approximated) representation of the dynamics of the MDP to estimate the entropy. However, the choice of whether to improve the policy using pure model free algorithms versus using the learned model is left as a design choice, beyond the scope of this work.

### 3.1 A Surrogate for Local Entropy

In conventional RL settings, one is able to sample rewards (and transitions, e.g. from a simulator). However, here, part of the expected reward to be maximized in equation 2 is the negative entropy $-\bar{h}^\pi$, which, as aforementioned, can be seen as an expected average reward with a state- and policy-dependent local reward $l^\pi(x)$. It is not reasonable to assume that one can directly sample local entropies $l^\pi(x)$: one would have to estimate $l^\pi(x)$ through estimating transition probabilities $P_\pi(x, y)$ (by sampling transitions) and using equation 3. However, a new challenge arises: $l^\pi$ *depends on the action distribution* and to apply average reward MDP theory we need the rewards to be state-action dependent. To address this, we consider a surrogate for $l^\pi$ that is policy-independent:

$$s(x, u) = -\sum_{y \in X} P(x, u, y) \log\left(P(x, u, y)\right).$$

Define $\bar{h}_s^\pi(x) := \lim_{T \to \infty} \mathbb{E}[\frac{1}{T} \sum_{t=0}^{T} s(X_t, \pi(X_t)) | X_0 = x]$. The following relationships hold.

**Lemma 1.** *Consider MDP $\mathcal{M} = (\mathcal{X}, \mathcal{U}, P, R, \mu_0)$. The following statements hold. a) $\mathbb{E}_{u \sim \pi(x)}[s(x, u)] \leq l^\pi(x)$, for all $\pi \in \Pi$. b) $\bar{h}_s^\pi(x) = \bar{h}_s^\pi \leq \bar{h}^\pi$, for some $\bar{h}_s^\pi \in \mathbb{R}$, for all $\pi \in \Pi$. c) $\mathbb{E}_{u \sim \pi(x)}[s(x, u)] = l^\pi(x)$ and $\bar{h}_s^\pi = \bar{h}^\pi$, for all $\pi \in \Pi^D$.*

Observe that, by considering the surrogate entropy rate, we effectively decouple the influence of the policy entropy in the entropy rate estimations. Policy stochasticity directly influences the true entropy rate $\bar{h}^\pi$, but does not affect the surrogate entropy rate $\bar{h}_s^\pi$; in a fully deterministic environment, $\bar{h}_s^\pi = 0$ for all $\pi \in \Pi$, but $\bar{h}^\pi$ would not. We make the case that, given the formal results in Lemma 1 (and the results to be presented in coming sections) this effect does not degrade the effectiveness of our method; in fact, it allows agents to find less uncertain environment regions while not directly discouraging exploration (and can still render minimum entropy rate policies, as discussed in Theorem 1 below).

### 3.2 Minimum Entropy Policies

Based on Lemma 1, we derive one of our main results. For the proof, we make use of fundamental results on existance of average reward optimal policies of MDPs. In particular Theorem 3 included in the Appendix, applied directly from Puterman (2014), which states that under mild assumptions the gain and bias exist in average reward MDPs for any policy, and that an optimal policy exists that maximises the reward rate.

**Theorem 1.** *Consider MDP $\mathcal{M} = (\mathcal{X}, \mathcal{U}, P, R, \mu_0)$. The following hold: a) There exists a deterministic policy $\hat{\pi} \in \Pi^D$ minimizing the surrogate entropy rate , i.e. $\hat{\pi} \in \arg\min_\pi \bar{h}_s^\pi$. b) Any $\hat{\pi} \in \Pi^D$ minimizing the surrogate entropy rate also minimizes the true entropy rate: $\hat{\pi} \in \arg\min_{\pi \in \Pi} \bar{h}_s^\pi$ and $\hat{\pi} \in \Pi^D$ $\implies$ $\hat{\pi} \in \arg\min_{\pi \in \Pi} \bar{h}^\pi$. Additionally, deterministic policies locally minimizing $\bar{h}_s^\pi$ also locally minimize $\bar{h}^\pi$. c) There exists a deterministic policy $\hat{\pi} \in \Pi^D$ such that $\hat{\pi} \in \arg\min_\pi \bar{h}^\pi$.*

*Proof of Theorem 1.* The first statement follows directly from Theorem 3 in the Appendix, which guarantees that there is at least one deterministic policy $\hat{\pi}$ that minimizes the surrogate entropy rate $\bar{h}_s^\pi$. Then, since $\hat{\pi} \in \Pi^D$, from Lemma 1 statements b) and c), we have that the following holds for all $\pi \in \Pi$:

$$\bar{h}^{\hat{\pi}} = \bar{h}^{\hat{\pi}} \leq \bar{h}_s^\pi \leq \bar{h}^\pi$$

Thus, $\hat{\pi}$ minimizes $\bar{h}^\pi$ and it follows that $\hat{\pi} \in \arg\min_{\pi \in \Pi} \bar{h}_s^\pi$ and $\hat{\pi} \in \Pi^D \implies \hat{\pi} \in \arg\min_{\pi \in \Pi} \bar{h}^\pi$. The same argument also applies locally, thereby yileding that deterministic local minimizers of $\bar{h}_s^\pi$ are also local minimizers of $\bar{h}^\pi$. Finally, the third statement follows as a combination of the other two. $\square$

Theorem 1 is an utterly relevant result for our work. First, it guarantees that minimizing policies both for $\bar{h}_s^\pi$ and $\bar{h}^\pi$ exist. More importantly, it tells us that, *to minimize the entropy rate of an RL agent, it is sufficient to minimize the surrogate entropy rate.* Since (globally) minimizing $\bar{h}_s^\pi$ implies minimizing $\bar{h}^\pi$ and since $s$ is policy-independent, in contrast to $l^\pi$, in what follows, our RL algorithm uses estimates of $s$ to minimize $\bar{h}_s^\pi$, instead of estimates of $l^\pi$ to minimize $\bar{h}^\pi$.[10]

---

[10]Observe that we cannot employ the same method for entropy rate maximization, since the maximizer of $\bar{h}_s^\pi$ is not necessarily a maximizer of $\bar{h}^\pi$.

## 4 Learning to Act Predictably

In the following, we show how predictability of the agent's behaviour can be cast as an RL objective and combined with a primary discounted reward goal. To do this, we rely on Theorem 1 and employ the surrogate entropy $s(x, u)$ as a local reward along with its corresponding value function. We prove that, given a learned model of the MDP, we are able to approximate the true entropy rate value functions. In the next section, we combine this section's results with conventional discounted rewards and standard PG results, to address the problem mentioned in the Problem Statement and derive a PG algorithm that maximizes the combined reward objective. We define the predictability objective to be minimized:

$$J_s(\pi) \equiv \bar{h}_s^\pi = \mathbb{E}\left[\lim_{T\to\infty} \frac{1}{T}\sum_{t=0}^{T} s\big(X_t, \pi(X_t)\big)\right].$$

Motivated by Theorem 1, we have employed the surrogate entropy as a local reward and consider the corresponding average-reward problem. As commonly done in average reward problems, we define the (surrogate) *entropy value function* for a policy $\pi$, $W^\pi : \mathcal{X} \to \mathbb{R}$ to be equal to the bias, i.e.:

$$W^\pi(x) := \mathbb{E}\left[\sum_{t=0}^{\infty} s\big(X_t, \pi(X_t)\big) - \bar{h}_s^\pi \mid X_0 = x\right] = \mathbb{E}_{y\sim P_\pi(x,\cdot)}\left[s\big(x, \pi(x)\big) - \bar{h}_s^\pi + W^\pi(y)\right], \qquad (6)$$

Additionally, we define the (surrogate) entropy action-value function $S^\pi : \mathcal{X} \times \mathcal{U} \to \mathbb{R}$ by $S^\pi(x, u) := \mathbb{E}_{y\sim P_\pi(x,u,\cdot)}\left[s\big(x, u\big) - \bar{h}_s^\pi + W^\pi(y)\right]$. However, recall that *we do not know* the local reward $s$. To estimate $s$, one needs to have an estimate of the transition function $P$ of the MDP. We use

$$s_\phi(x, u) = -\sum_{y\in\mathcal{X}} P_\phi(x, u, y) \log\big(P_\phi(x, u, y)\big)$$

(and $\bar{h}_{s_\phi}^\pi$ correspondingly, for its associated rate) to denote the – parameterised by $\phi$ – estimate of $s$, which results from a corresponding estimate $P_\phi$ of $P$ (i.e. $P_\phi$ is the learned model). Similarly, we will use $J_{s_\phi}$, $W_\phi^\pi$ and $S_\phi^\pi$ to denote value functions computed with the model estimates $s_\phi$. Now, it is crucial to know that by using the model estimates $s_\phi$ we are still able to approximate well the objective $J_s$ and the value functions $W^\pi, S^\pi$. Let us first show that for a small error between $P_\phi$ and $P$ (i.e. small modeling error), the error between $s_\phi$ and $s$ and the objectives $J_s(\pi) \equiv \bar{h}_s^\pi$ and $J_{s_\phi}(\pi) \equiv \bar{h}_{s_\phi}^\pi$ is also small.

**Proposition 1.** *Consider MDP $\mathcal{M} = (\mathcal{X}, \mathcal{U}, P, R, \mu_0)$. Consider $P_\phi : \mathcal{X} \times \mathcal{U} \times \mathcal{X} \to [0, 1]$, parameterised by $\phi \in \Phi$. Assume that the total variation error between $P_\phi$ and $P$ is bounded as, $\forall\ x \in \mathcal{X}$ and $u \in \mathcal{U}$,* $\max_{x\in\mathcal{X}, u\in\mathcal{U}} D_{TV}\big(P_\phi(x, u, \cdot) \| P(x, u, \cdot)\big) \leq \epsilon$, *for some $\epsilon$, with $0 \leq \epsilon \leq 1$. Let $K(\epsilon) = \epsilon \log\big(|\mathcal{X}|\big) - \epsilon \log \epsilon$. Then, $\|s_\phi(x, u) - s(x, u)\|_\infty \leq K(\epsilon)$, and the surrogate entropy rate error for any policy $\pi$,*

$$\mathbb{E}\big[\lim_{T\to\infty} \frac{1}{T}\sum_{t=0}^{T} s_\phi(X_t, \pi(X_t)) \mid X_0 \sim \mu_0\big] - \bar{h}_s^\pi \leq K(\epsilon).$$

The proof of proposition 1 hinges on the Fannes-Audenaert inequality for Von Neumann entropies Fannes (1973); Audenaert (2007), where we simply assume the density matrices are diagonal matrices with the transition probability densities as eigenvalues. Observe that as $\epsilon \to 0$, i.e. as the learned model approaches the real one, then the surrogate entropy rate converges to the actual one[11] (since $K(\epsilon) \to 0$). This result indicates that we can indeed use $s_\phi$, obtained by a learned model $P_\phi$, instead of the unknown $s$, as the error between the objectives $J_s(\pi) \equiv \bar{h}_s^\pi$ and $J_{s_\phi}(\pi) \equiv \bar{h}_{s_\phi}^\pi$ is small, for small model errors. Assume now, without loss of generality that we have parameterised entropy value function (*critic*) $S_\omega$ with parameters $\omega \in \Omega$. We show that a standard on-policy algorithm, with policy $\pi$, with value function approximation $S_\omega$, using the approximated model $P_\phi$, learns entropy value functions that are in a $\delta(\epsilon)$-neighbourhood of the true entropy value functions $S^\pi$, and $\delta(\epsilon)$ vanishes with $\epsilon$.

---

[11] This result echoes the Simulation Lemma in Kearns & Singh (2002), but with a bound derived in infinite horizon by using the entropy properties.

**Assumption 1.** *Any learning rate $\alpha_t \in (0,1)$ satisfies $\sum_{t=0}^{\infty} \alpha_t = \infty$, $\sum_{t=0}^{\infty} \alpha_t^2 < \infty$.*

**Assumption 2.** *The model $P_\phi$ satisfies $\max_{x \in \mathcal{X}, u \in \mathcal{U}} D_{TV}(P_\phi(x,u,\cdot) \| P(x,u,\cdot)) \le \epsilon$ for a small $\epsilon \in [0,1]$.*

**Proposition 2.** *Consider an MDP $\mathcal{M}$, a policy $\pi$, a learned model $P_\phi$ of the MDP and critic $S_\omega$ linear on $\omega$, and $\omega \in \Omega \subset \mathbb{R}^n$, where $\Omega$ is compact. Let Assumption 2 hold. At every step $k$ of parameter iteration, let us collect $k$ trajectories $\mathcal{T}_k$ of length $T$, and construct (unbiased) estimates[12] $\hat{S}_\phi^\pi$. Let the critic parameters $\omega \in \Omega$ be updated as $\omega_{k+1} = \omega_k - \beta_k \hat{\Delta\omega}_k$, with $\omega_0 \in \Omega$, $\beta_k$ being a learning rate and*

$$\hat{\Delta\omega}_k = \left( \hat{S}_\phi^\pi(x_k, u_k) - S_\omega(x_k, u_k) \right) \frac{\partial S_\omega(x_k, u_k)}{\partial \omega}.$$

*Then, $\omega$ converge to a $\delta(\epsilon)$-neighbourhood of one of the (local) minimizers of $\mathbb{E}_{\substack{x \sim \mu^\pi \\ u \sim \pi_\theta(x)}} \left[ \frac{1}{2} \left( S^\pi(x,u) - S_\omega(x,u) \right)^2 \right]$, where $\delta(\epsilon)$ is vanishing with $\epsilon$.*

In other words, for small model errors, the value function approximator converges to a locally optimal value function approximation of the true value function $S^\pi$.

## 4.1 Predictability-Aware Policy Gradient

Now, we are ready to address the Problem Statement, combining the entropy rate objective with a discounted reward objective. In what follows, assume that we have a parameterised policy $\pi_\theta$ with parameters $\theta \in \Theta$. Let $J(\pi_\theta) = \mathbb{E}[\sum_{t=0}^{\infty} \gamma^t R_t^{\pi_\theta}]$. We use $Q_\xi$ with parameters $\xi \in \Xi$ for the parameterised critic of the discounted reward objective (when using a form of actor-critic algorithm).

**Theorem 2.** *Consider an MDP $\mathcal{M}$, parameterised policy $\pi_\theta$, a learned model $P_\phi$ of the MDP and (linear) critic $S_\omega$. Let Assumption 2 hold. Let a given PG algorithm maximize (locally) the discounted reward objective $J(\pi_\theta) = \mathbb{E}[\sum_{t=0}^{\infty} \gamma^t R_t^{\pi_\theta}]$. Let the value function $Q_\xi$ (or $V_\xi$) be parameterised by $\xi \in \Xi$, and the entropy value function $S_\omega$ (or $W_\omega$) have the same parameterisation class. Then, the same PG algorithm with updates*

$$\theta \leftarrow \text{proj}_\Theta \left[ \theta + \alpha_t \left( \hat{\nabla}_\theta J(\pi_\theta) - k \hat{\nabla}_\theta J_{s_\phi}(\pi_\theta) \right) \right]$$

*converges to a local maximum of the combined objective $J(\pi_\theta) - k J_{s_\phi}(\pi_\theta)$.*

*Proof of Theorem 2 (Sketch).* By standard PG arguments Sutton et al. (1999), if a PG algorithm converges to a local maximum of the objective $J(\pi_\theta)$ then the updates $\hat{\nabla}_\theta J(\pi_\theta)$ are in the direction of the gradient (up to stochastic approximation noise). By the same arguments, given Proposition 2, the same algorithm converges to a local minimum of the entropy value function $W_\omega$ through updates $-\hat{\nabla}_\theta J_s(\pi_\theta)$, and these are in the direction of the true gradient (again, up to stochastic approximation noise). Then, the linear combination of gradient updates $\hat{\nabla}_\theta J(\pi_\theta) - k \hat{\nabla}_\theta J_s(\pi_\theta)$ is in the direction of the gradient of the combined objective $J(\pi_\theta) - k J_s(\pi_\theta)$. Finally, since both objectives are locally concave (necessary condition following from existence of gradient schemes that locally maximize them), their linear combination is also locally concave. This concludes the proof. □

**Remark 1.** *Regarding pure entropy rate minimization, i.e. without the discounted reward objective, as already proven by Theorem 1, a policy that is globally optimal for the surrogate entropy rate $J_s(\pi)$ is also optimal for the actual entropy rate $\bar{h}^\pi$. The same holds for locally optimal deterministic policies. However, in general, this is not the case for stochastic local minimizers.*

Following a vanilla policy gradient structure, in Algorithm 1 we first sample a trajectory $\tau$ of length $T$, under a policy $\pi_\theta$, and store it in a buffer $\mathcal{D}$ (for training the approximate model $P_\phi$). Then, we use $\mathcal{D}$ to train $P_\phi$; update the estimated entropy rate; compute estimated objective gradients $\hat{\nabla}_\theta J(\pi_\theta)$, $\hat{\nabla}_\theta J_{s_\phi}(\pi_\theta)$ from trajectory $\tau$[13]; and finally update the policy and critics $S_\omega, Q_\xi$.

**Remark 2.** *If we were to consider average reward MDPs instead of discounted reward MDPs, the formulation of the optimization problem solved in Theorem 2 results in a more natural interpretation when adding entropy rate objectives. See Appendix 6 for details.*

---

[12]Via *e.g.* TD(0) value estimation Sutton & Barto (2018).

[13]This can be done through any policy gradient algorithm at choice.

### 4.2 Implementation

**Policy Learning**  We implement our predictability-aware scheme as first, an on-policy version based on an average-reward PPO algorithm Ma et al. (2021); we call this PAPPO. In particular, for every collected trajectory $\tau$, we update the estimate $\bar{h}^\pi_{s_\phi}$ and (surrogate) entropy value function $W_\omega$ parameterised by $\omega \in \Omega$ from the collected samples, and we compute *entropy advantages* for all $(x, u, y, s_\phi(x, u))$ in the collected trajectories as: $\hat{A}^\pi_s = s_\phi(x,u) - \bar{h}^\pi_{s_\phi} + W_\omega(y) - W_\omega(x)$. Then we apply the gradient steps as in PPO Schulman et al. (2015; 2017) by clipping the policy updates. Second, we implement it in an off-policy fashion to compare with recent results on information-theoretic RL, based on a Soft Actor-Critic (Haarnoja et al., 2018) implementa-

---

**Algorithm 1** Predictability Aware Policy Gradient

---

**Require:** $P_\phi, \pi_\theta$, critics $W_\omega, V_\xi$
**Require:** $\alpha_t, k > 0$
  **for** $E$ epochs **do**
    $\mathcal{D} \leftarrow$ Trajectory $\tau$ of length $T$.
    Train $P_\phi$ from $\mathcal{D}$.
    $\bar{h}^\pi_{s_\phi} \leftarrow \frac{1}{T} \sum_{x,u \in \tau} s_\phi(x, u)$.
    Compute $\hat{\nabla}_\theta J(\pi_\theta), \hat{\nabla}_\theta J_{s_\phi}(\pi_\theta)$ from $\tau$.
    $\theta \leftarrow \mathrm{proj}_\Theta \left[ \theta + \alpha_t \left( \hat{\nabla}_\theta J(\pi_\theta) - k \hat{\nabla}_\theta J_{s_\phi}(\pi_\theta) \right) \right]$
    Update $S_\omega$ (and $Q_\xi$ if used)
  **end for**

---

tion; we call this PASAC. The only modification necessary is the learning of a second $Q$ function ($S$) for the surrogate trajectory entropy, and the policy loss is computed as a weighted sum of $Q$ and $S$. See Appendix C.1 for details on PASAC.

**Model Learning**  To learn the approximated model $P_\phi$, we assume the transitions to follow Gaussian distributions, similarly to Janner et al. (2019).[14] Following the definition of *differential entropy* of a continuous Gaussian distribution, $s_\phi(x, u) = \log(\sigma^2_{xu}) + K$, where $K = \frac{1}{2}(\log(2\pi) + 1)$. Therefore, we can estimate the entropy directly by the variance output of our model. Furthermore, since we only need to estimate the variance per transition $(x, u) \rightarrow y$, it is sufficient to construct a model $f_\phi : \mathcal{X} \times \mathcal{U} \rightarrow \mathcal{X}$ that approximates the *mean* $f_\phi(x, u) \approx \int y P(x, u, y) dy$, and we do this through minimizing a mean-squared error loss of transition samples in our model. Then, we estimate the entropy of an observed transition as $s_\phi(x, u) = \log(\mathbb{E}_{y \sim P(x, u, \cdot)}[(f_\phi(x, u) - y)^2])$.

**Entropy Rate Computation**  Given a trained agent, the entropy rates are estimated equally for all algorithms. At inference (when rolling out the trained agent), given the trained model $f_\phi$, the entropy rate for a trajectory $\{(X_t, \pi(X_t))\}_{0 \le t \le T-1}$ of length $T$ is estimated as $\bar{h}^\pi = \frac{1}{T} \sum_{t=0}^{T-2} \log((f_\phi(X_t, \pi(X_t)) - X_{t+1})^2])$.

## 5 Experiments

We implemented PARL on a set of robotics and autonomous driving tasks, evaluated the obtained rewards and entropy rates, and compared against different baselines. For the MuJoCo hyperparameters, we took pre-tuned values from Raffin et al. (2019). For the experiments using PASAC and explicit comparison against other SAC-based baselines including RPC (Eysenbach et al. 2021), see Appendix C.1. [15]

**Rewards, Entropies and Ablation**  To evaluate the influence of the trade-off parameter $k$, we test PAPPO on MuJoCo tasks and compare to on-policy baselines. We train all agents using the same hyperparameters, and we only vary the trade-off $k$ in the PARL agents to evaluate the influence. Additionally, we run both the deterministic and stochastic resulting policies (deterministic chooses the mean action that comes out of the policy, stochastic samples from it). The results are reported in Figure 2. Note that, as Mujoco tasks have continuous state and action spaces, entropy rates may be negative.[16]

---

[14]This is a strong assumption, and in many multi-modal problems, it may not be sufficient to capture the dynamics. Note, however, that our method is compatible with any representation of learned model $P_\phi$.

[15]We have also designed two additional representative robotic tasks where agents use PARL to avoid unnecessary stochasticity in the environment. See the Appendix for these.

[16]Recall that, for our implementation, to address continuous state/action tasks, we use *differential entropy* to quantify predictability for continuous random variables, which may indeed be negative. A different metric one may use is *relative entropy* (the KL-divergence from the uniform distribution).

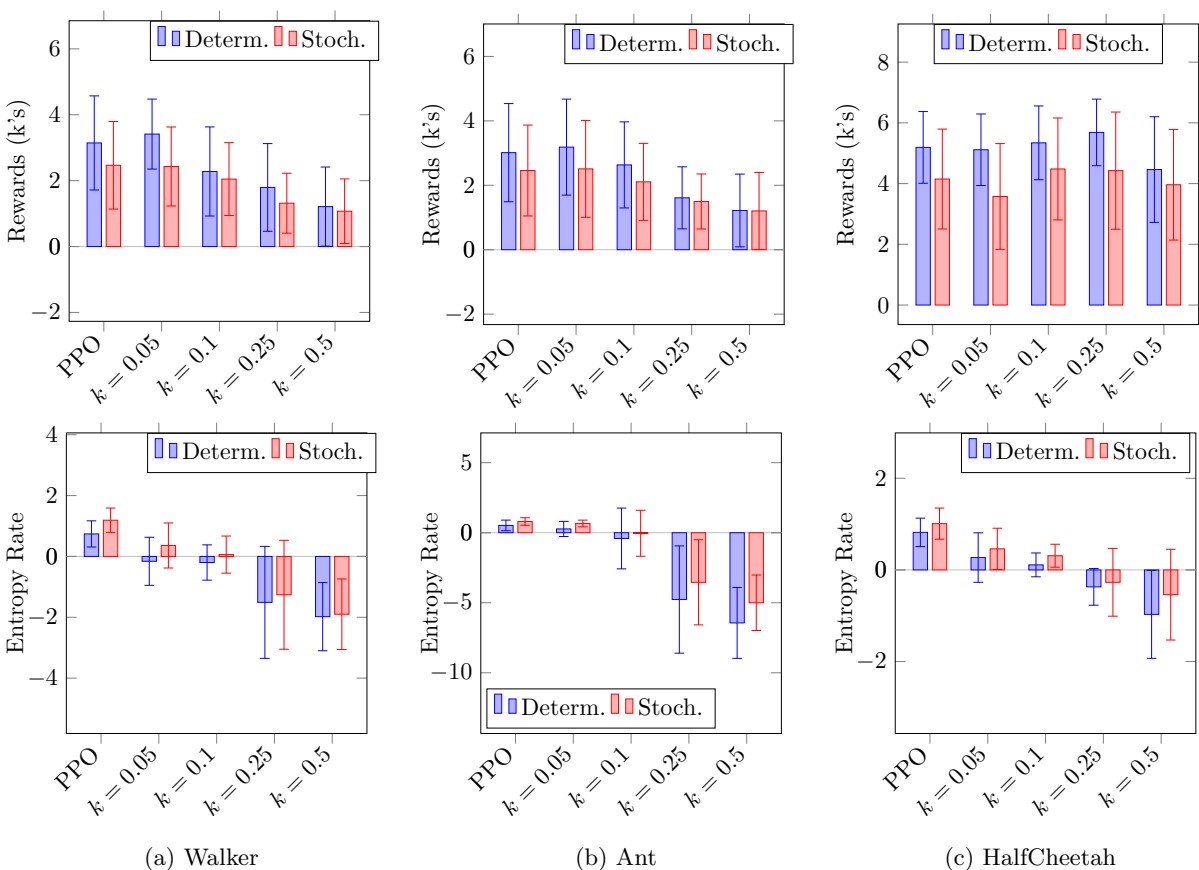

Figure 2: Rewards and Entropy rates for PAPPO as a function of $k$. Leftmost is PPO baseline.

**Trajectory Complexity in MuJoCo**  The effect of entropy-rate minimization and PAPPO on trajectory distributions is showcased in Figure 3. We evaluated trained agents over 10 full episodes, and plotted the observed trajectories in task space to compare trajectory distributions. We plot as representative variables the $z$-position and angle of the front tip of the robot. In both cases, we observe that PAPPO policies induce more regular, clustered trajectories, which suggests smaller distributional complexity. Generally, PAPPO trajectories have considerably smaller variance, and visit a smaller portion of the state-space. In the Walker and Ant environments this difference is very pronounced, as PAPPO trajectories are concentrated in a tiny region of the state-space, and especially in terms of the angles observed, are limited to a much smaller range.

**Predictable Driving**  We test PAPPO in the Highway Environment (Leurent, 2018), where an agent learns to drive at a desired speed while navigating a crowded road with other autonomous agents. The agent gets rewarded for tracking the desired speed and penalised for collisions. We consider a *highway* and a *roundabout* scenario (see Figure 4). We compare against PPO (Schulman et al., 2017) and DQN Mnih et al. (2015)) agents, and take the hyperparameters directly from Leurent (2018). The results are presented in Table 1. In the highway environment, agents slow down their speed and stop overtaking (arguably, a more predictable driving pattern). This results in longer episode lengths (and larger episodic rewards), but lower rewards per time-step (since reward is given for driving faster). In the roundabout scenario, a different behaviour emerges: Agents keep a constant high speed to traverse the roundabout as fast as possible, as the roundabout is the main source of complexity.

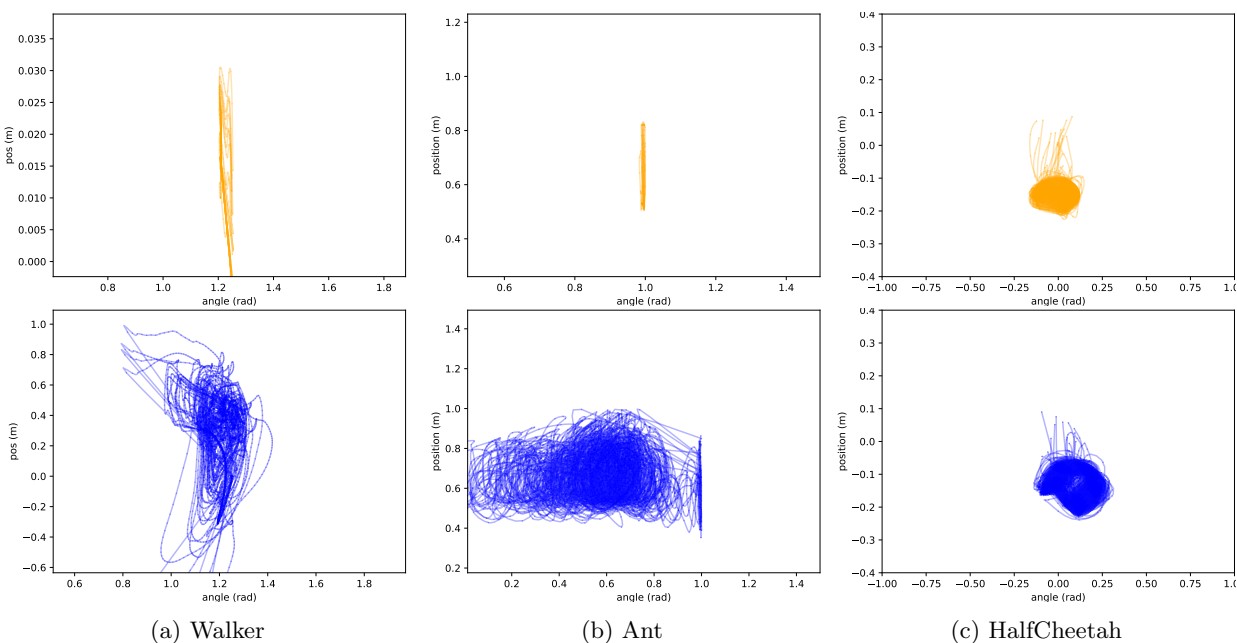

Figure 3: 10 trajectory plots for agents with seed 0. Blue is PPO, orange is PAPPO with $k = 0.5$.

Table 1: Results for autonomous driving environments.

| Highway | Rewards | Ep. Length | Avg. Speed | Entropy Rate |
|---|---|---|---|---|
| DQN | $17.51 \pm 7.73$ | $20.92 \pm 9.08$ | $\mathbf{6.51 \pm 0.34}$ | $-0.37 \pm 0.16$ |
| PPO | $18.88 \pm 6.74$ | $24.41 \pm 8.49$ | $5.62 \pm 0.10$ | $-0.88 \pm 0.05$ |
| $\text{PAPPO}_{k=0.1}$ | $\mathbf{21.05 \pm 4.03}$ | $28.56 \pm 5.27$ | $5.11 \pm 0.05$ | $-1.43 \pm 0.09$ |
| $\text{PAPPO}_{k=0.5}$ | $21.03 \pm 2.05$ | $29.60 \pm 2.64$ | $5.06 \pm 0.01$ | $-1.51 \pm 0.09$ |
| $\text{PAPPO}_{k=1}$ | $20.89 \pm 2.05$ | $29.61 \pm 2.58$ | $5.05 \pm 0.00$ | $\mathbf{-1.51 \pm 0.06}$ |

| Roundabout | Rewards | Ep. Length | Avg. Speed | Entropy Rate |
|---|---|---|---|---|
| DQN | $22.53 \pm 11.93$ | $26.92 \pm 14.41$ | $3.09 \pm 0.47$ | $-0.55 \pm 0.88$ |
| PPO | $\mathbf{29.26 \pm 10.68}$ | $32.92 \pm 11.67$ | $2.80 \pm 0.12$ | $-0.23 \pm 0.27$ |
| $\text{PAPPO}_{k=0.1}$ | $28.86 \pm 10.96$ | $32.98 \pm 12.23$ | $2.65 \pm 0.50$ | $-0.28 \pm 0.77$ |
| $\text{PAPPO}_{k=0.5}$ | $17.99 \pm 13.34$ | $23.45 \pm 17.60$ | $3.75 \pm 0.05$ | $-2.13 \pm 0.12$ |
| $\text{PAPPO}_{k=1}$ | $16.83 \pm 13.28$ | $22.05 \pm 17.67$ | $\mathbf{3.79 \pm 0.04}$ | $\mathbf{-2.21 \pm 0.14}$ |

## 6 Average Reward Formulation

Through the work we argue that the entropy rate can be efficiently cast and implemented as an average reward criterion, to be combined with other primary reward objectives. A natural question to ask is how does the formulation of our work adapt to the case where the primary objective is an average reward objective over the MDP rewards. Consider the case where we are interested in optimizing a linear combination of average reward and entropy rate objectives. Then, we can write the reward objective as $J(\pi) = \mathbb{E}\left[\lim_{T\to\infty} \frac{1}{T} \sum_{t=0}^{T} R_t^\pi\right]$, and since both the average rewards and the average entropies are taken in expectation over the same probability space, the trade-off objective in equation 2 becomes $\arg\max_{\pi \in \Pi} J(\pi) - k J_s(\pi) = \mathbb{E}\left[\lim_{T\to\infty} \frac{1}{T} \sum_{t=0}^{T} R_t^\pi - k s_t^\pi\right]$, where we already included the surrogate entropy $s_t^\pi = s(X_t, \pi(X_t))$. Further, recall that $s(x, u) = \mathbb{E}_{y\sim P(x,u,\cdot)}[-\log P(x, u, y)]$. Then, assuming knowledge of $P$ (or an approximation of it), one can define an adjusted reward $\tilde{R}(x, u, y) := R(x, u, y) + k \log P(x, u, y)$ and a modified value function $\tilde{V}_{\text{avg}}^\pi(x) := \mathbb{E}_{u\sim\pi, y\sim P(x,u,\cdot)}[R(x, u, y) + k \log P(x, u, y) - \tilde{g}^\pi + V_{\text{avg}}^\pi(y)]$, where $\tilde{g}^\pi = \mathbb{E}\left[\lim_{T\to\infty} \frac{1}{T} \sum_{t=0}^{T} \tilde{R}_t^\pi\right]$ is the modified reward rate under policy $\pi$ (which is independent of the initial set of states for ergodic MDPs). Observe that, even though $\tilde{R}(x, u, y) \neq R(x, u, y) - k s(x, u)$, they are equal in expectation. Then, for any fixed policy, the expected average reward can be computed through $\tilde{R}$, and a

policy $\pi^*$ solving the optimization problem $\pi^* \in \arg\max_{\pi \in \Pi} \tilde{V}^\pi_{\text{avg}}(x) \quad \forall x \in \mathcal{X}$ is guaranteed to solve the average reward objective. This allows for more compact formulation, and for the learning of a single value function (instead of two), which can be desirable in some cases. It also implies that transitions with low probabilities of being observed are penalized. Computationally, this is also advantageous since we do not need to compute the entropy of the learned model, but instead evaluate the likelihood of the observed transition and adjust the rewards accordingly.

## 7 Discussion

**Summary of Results** We proposed a novel method, namely PARL, that induces more predictable behaviour in RL agents by maximizing a tuneable linear combination of a standard expected reward and the negative entropy rate, thus trading-off *optimality with predictability.* In the experimental results, we see how PARL greatly reduces the entropy rates of the RL agents while achieving near-optimal rewards, depending on the trade-off parameter. In the autonomous driving setting, agents learn to be more predictable while driving around stochastic agents. In the MuJoCo experiments, PARL obtains policies that yield more clustered, less complex trajectory distributions, allowing models to predict better the dynamics. This results in, for example, a smaller range of values of orientation angles as seen in the trajectory representations on Figure 3. Additionally, following our method, the entropy rate can be directly interpreted as the average complexity necessary to correctly predict the trajectory of the agents, and if assuming Gaussian predictions, this is proportional to the log of the prediction variance observed by the agent's internal prediction model, which shows how lower entropy rates yield predictability.

**Shortcomings** Our scheme results in a setting where agents maximize a trade-off between two different objectives. This, combined with learning a dynamic model (which is notoriously difficult), introduces implementation challenges related to learning multiple coupled models simultaneously. We make an attempt at discussing a systematic way of addressing these in the Appendix. Additionally, for many applications, avoiding high-entropy policies may restrict the ability of RL agents to learn optimal behaviours (see, e.g., results in Eysenbach et al. 2021). And although in human-robot interaction and human-aligned AI predictability is intuitively beneficial, we cannot claim that minimizing entropy-rates is desirable for all RL applications. Additionally, a question remains regarding how would our method change in partially observable environments. The entropy rate of a POMDP (if measured on the state sequence, not on observations) is influenced by the policy given an observation, observation probability given a state, and the state transition, resulting in a much more difficult object to estimate (see e.g. Savas et al. (2022)). We believe the theoretical and practical considerations of minimising entropy rates in POMDPs (in model-free settings) will be a very interesting future direction.

**Entropy and Exploration** One might wonder if minimizing entropy rates of RL agents may hinder exploration. From a theoretical perspective, this is not a problem due to ergodicity. From a practical perspective, this undesired effect is highly mitigated by the fact that for the first (many) steps, the agent is still learning an adequate model of the dynamics, and therefore the entropy signal is very noisy which favors exploration.

## Acknowledgements

Authors want to thank Alvaro Serra, Lasse Peters, Khaled A. Mustafa and Elia Trevisan for the useful discussions on this topic. This research was supported by funding from the Dutch Research Council NWO-NWA, within the "Acting under uncertainty" (ACT) project (Grant No. NWA.1292.19.298).

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

## A   Auxiliary Results

We include in this Appendix some existing results used throughout our work. The following Theorem is a combination of results presented by Puterman (2014) regarding the existence of optimal average reward policies.

**Theorem 3** (Average Reward Policies Puterman (2014)). *Given an MDP $\mathcal{M}$ the following hold: $g^\pi(x)$ and $b^\pi(x)$ exist, $g^\pi(x) = g^\pi$ for all $x$ (i.e. $g^\pi(x)$ is constant), and there exists a deterministic, stationary policy $\hat{\pi} \in \arg\max_{\pi \in \Pi} g^\pi$ that maximizes the expected average reward. Additionally, the same holds if $\mathcal{U}$ is compact and $R$ and $P$ are continuous functions of $\mathcal{U}$.*

**Theorem 4** (Stochastic Recursive Inclusions Borkar (2009)). *Let $x_n \in \mathbb{R}^d$ be a vector following a sequence:*

$$x_{n+1} = x_n + a_n(h(x_n) + M_{n+1} + \eta_n), \tag{7}$$

*where $\sup_n \|x_n\| < \infty$ a.s., $a_n$ is a learning rate, $h : \mathbb{R}^d \to \mathbb{R}^d$ is a Lipschitz map, $M_n$ is a Martingale difference sequence with respect to the $\sigma-$algebra $\mathcal{F}_n := \sigma(x_0, M_1, M_2, ..., M_n)$ (and square integrable), and $\eta_n$ is an error term bounded by $\epsilon_0$. Define $H := \{x \in \mathbb{R}^d : h(x) = 0\}$. Then, for any $\delta > 0$, there exists an $\epsilon > 0$ such that $\forall \epsilon_0 \in (0, \epsilon)$ the sequence $\{x_n\}$ converges almost surely to a $\delta$-neighbourhood of $H$.*

**Theorem 5** (Policy Gradient with Function Approximation Sutton et al. (1999)). *Let $\pi_\theta$ be a parameterised policy and $f_w : \mathcal{X} \times \mathcal{U} \to \mathbb{R}$ be a parameterised (approximation of) action value function in an MDP $\mathcal{M}$. Let the parameterisation be compatible, i.e. satisfy:*

$$\frac{\partial f_w(x, u)}{\partial w} = \frac{\partial \pi_\theta(x, u)}{\partial \theta} \frac{1}{\pi_\theta(x, u)}.$$

*Let the parameters $w$ and $\theta$ be updated at each step such that:*

$$w_k : \sum_x \mu^{\pi_\theta}(x) \sum_u \pi_\theta(x, u) \big( Q^{\pi_\theta}(x, u) - f_{w_k}(x, u) \big) \frac{\partial f_w(x, u)}{\partial w_k} = 0,$$

$$\theta_{k+1} = \theta_k + \alpha_t \sum_x \mu^{\pi_\theta}(x) \sum_u \frac{\partial \pi_\theta(x, u)}{\partial \theta} f_{w_k}(x, u).$$

*Then, $\lim_{k \to \infty} \frac{\partial \rho(\theta_k)}{\partial \theta} = 0$, where $\rho(\theta)$ is either the discounted or average reward in the MDP.*

## B   Technical Proofs

*Proof of Lemma 1.* For statement $a$), observe $l^\pi(x)$ can be expressed as

$$l^\pi(x) = -\mathbb{E}\left[P(x, u, y)|u \sim \pi(x)\right] \log\left(\mathbb{E}\left[P(x, u, y)|u \sim \pi(x)\right]\right),$$

and recall

$$\mathbb{E}_{u \sim \pi(x)}[s(x, u)] = -\mathbb{E}_{u \sim \pi(x)}\Big[ \sum_{y \in X} P(x, u, y) \log\left(P(x, u, y)\right) \Big].$$

Then, from Jensen's inequality, $\mathbb{E}_{u \sim \pi(x)}[s(x, u)] \leq l^\pi$.

In Statement $b$), the fact that $\bar{h}_s^\pi(x)$ is constant follows from Theorem 3 by considering $R(x, u, y) \equiv s(x, u)$. Now, note that we can write $\bar{h}_s^\pi = \sum_{x \in \mathcal{X}} \mathbb{E}_{u \sim \pi(x)}[s(x, u)]\mu^\pi(x)$ and $\bar{h}^\pi = \sum_{x \in \mathcal{X}} l^\pi(x)\mu^\pi(x)$ (see Puterman (2014)). Thus:

$$\bar{h}_s^\pi = \sum_{x \in \mathcal{X}} \mathbb{E}_{u \sim \pi(x)}[s(x, u)]\mu^\pi(x) \leq \sum_{x \in \mathcal{X}} l^\pi(x)\mu^\pi(x) = \bar{h}^\pi,$$

where we employed Statement $a$).

For Statement $c$), take $\pi \in \Pi^D$. Then, $\pi(u' \mid x) = 1$ and $\pi(u \mid x) = 0$ for an action $u' \in \mathcal{U}$ and all $u \neq u'$. Then

$$\mathbb{E}_{u \sim \pi(x)}[s(x, u)] = \sum_{y \in X} P(x, u', y) \log\left(P(x, u', y)\right) = l^\pi(x).$$

The fact that $\bar{h}_s^\pi = \bar{h}^\pi$ follows, then, trivially. $\qquad\square$

*Proof of Proposition 1.* Observe that $s(x, u)$ and $s_\phi(x, u)$ are the entropies of probability distributions $P(x, u, \cdot)$ and $P_\phi(x, u, \cdot)$, respectively. Thus, the Fannes–Audenaert inequality Audenaert (2007) for two probability distributions $p$ and $q$ states:

$$|H(p) - H(q)| \leq 2T \log(d) - 2T \log(2T),$$

where $T = \frac{|p-q|_1}{2}$ and $d$ is the dimensions of the support of the distributions. Then, taking $P(x, u, \cdot)$ and $P_\phi(x, u, \cdot)$ as distributions, for any action $u$ we obtain:

$$\|s_\phi(x, u) - s(x, u)\|_\infty \leq \epsilon \log \left( |\mathcal{X}| \right) - \epsilon \log \epsilon =: K(\epsilon),$$

Finally:

$$\mathbb{E}\big[ \lim_{T \to \infty} \frac{1}{T} \sum_{t=0}^{T} s_\phi(X_t, \pi(X_t)) \mid X_0 \sim \mu_0 \big] - \bar{h}_s^\pi =$$

$$\mathbb{E}\big[ \lim_{T \to \infty} \frac{1}{T} \sum_{t=0}^{T} s_\phi(X_t, \pi(X_t)) - s(X_t, \pi(X_t)) \big] \leq$$

$$\mathbb{E}\Big[ \lim_{T \to \infty} \frac{1}{T} \sum_{t=0}^{T} \big| s_\phi(X_t, \pi(X_t)) - s(X_t, \pi(X_t)) \big| \Big] \leq \mathbb{E}\big[ \lim_{T \to \infty} \frac{1}{T} \sum_{t=0}^{T} K(\epsilon) \big] = K(\epsilon).$$

$\square$

*Proof of Proposition 2.* Since $S_\omega$ is linear on $\omega \in \Omega$ and $\Omega$ is compact, there exists at least one minimizer $\omega^*$. Now, from equation 1 and Theorem 8.2.6 (Puterman, 2014), $S^\pi$ and $S_\phi^\pi$ can be written in vector form (over the states $\mathcal{X}$) as:

$$S^\pi = (I - P_\pi + P_\pi^*)^{-1}(I - P_\pi^*)s^\pi,$$
$$S_\phi^\pi = (I - P_\pi + P_\pi^*)^{-1}(I - P_\pi^*)s_\phi^\pi,$$

where $s^\pi$ is the vector representation of $s(\cdot, \pi(\cdot))$ (and analogously for $s_\phi^\pi$). Therefore, from Proposition 1, $\|S^\pi(x) - S_\phi^\pi(x)\|_\infty \leq K(\epsilon)$. Then, we can write without loss of generality

$$S^\pi(x, u) = S_\phi^\pi(x, u) + \eta(\epsilon, x),$$

with $\eta(\epsilon, x)$ being $O(\epsilon)$ for all $x$ since it is bounded by a function $K(\epsilon$ which is $O(\epsilon)$. Finally, we can write the parameter iteration as

$$\omega_{t+1} = \omega_t + \beta_t \big[ -\nabla_\omega L_\omega^\pi + M_{t+1} + \eta(\epsilon) \big],$$

with $L_\omega^\pi := \mathbb{E}_{\substack{x \sim \mu^\pi \\ u \sim \pi_\theta(x)}} \left[ \frac{1}{2} \left( S^\pi(x, u) - S_\omega(x, u) \right)^2 \right]$ and the term $M_{t+1} := \left( \hat{S}^\pi(x, u) - S^\pi(x, u) \right) \frac{\partial S_\omega(x, u)}{\partial \omega}$ is a Martingale with bounded variance (since $s$ is bounded).

Therefore, by Theorem 6 in (Borkar, 2009), the iterates converge to some point $\omega_t \to \Omega_\delta^*(\pi_\theta)$ *almost surely* as $t \to \infty$, with $\Omega_\delta^*(\pi_\theta)$ being the $O(\delta)$ neighbourhood of the stationary points satisfying $\nabla_\omega L_\omega^\pi = 0$. $\square$

## C   Experimental Results and Methodology

We present here the extended experimental results, training curves and additional details corresponding to the experimental framework. All experiments were run in a single CPU, running Ubuntu 20.04, and all libraries and requirements are properly listed in the paper code.

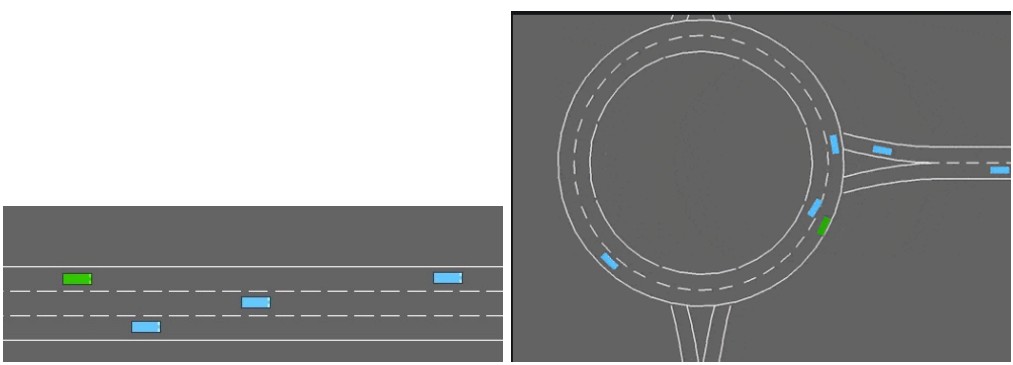

Figure 4: Self-Driving Environments.

## C.1 Soft Actor-Critic Experiments

As discussed in Section 4.2, we implemented a version of PARL based on a Soft Actor-Critic implementation (Haarnoja et al., 2018). For this, we simply learn in parallel an average reward Q function to optimize the entropy rate ($S_\omega$), and combine the Q functions to compute the actor objective as:

$$J_{sac}(\pi) = \mathbb{E}_{x \sim \mathcal{D}} \left[ \mathbb{E}_{u \sim \pi} \left[ \alpha \log \pi(u \mid x) - Q_\xi(x, u) + \\ + k S_\omega(x, u) \right] \right]. \tag{8}$$

As a baseline for our SAC implementation, we use RPC (Eysenbach et al., 2021), which is another SAC based algorithm aiming to maximize rewards while compressing policies to a maximum complexity, to achieve more simple, robust and predictable behaviours. Please note, the comparison is merely *qualitative*: RPC does not optimize for entropy rate in the agent's behaviour, however simpler policies do induce smaller entropy rates (as seen in the reported results) and thus it is still useful as a baseline to evaluate the impact of entropy rate objectives in SAC agents. We trained all agents with the same parameters, 5 agents per parameter combination and evaluated over 50 independent episodes. For RPC, we trained agents with different policy compression rates (2 bits and 5 bits), and for PASAC with two different trade-off parameters ($k = 1$ and $k = 0.5$). We train both PASAC and RPC with the same architecture for the prediction models (*decoder* in RPC), and we use an identity encoder for RPC to make the prediction models equivalent, and the entropy to be estimated in task space (not in a latent space). The results are presented in table 2. We also include a DDPG () baseline[17] which is an inherently deterministic policy.

Table 2: Results for MuJoCo environments, Off Policy algorithms.

| Ant | Rewards | Ep. Length | Entropy Rate |
|---|---|---|---|
| DDPG | **1371.23 $\pm$ 757.25** | 899.60 $\pm$ 170.73 | 4.14 $\pm$ 0.91 |
| PASAC$_{k=1}$ | **6173.43 $\pm$ 442.87** | 997.12 $\pm$ 48.45 | **-0.70 $\pm$ 0.12** |
| PASAC$_{k=0.5}$ | 5235.99 $\pm$ 1268.48 | 957.72 $\pm$ 168.55 | -0.27 $\pm$ 0.44 |
| RPC$_{2bit}$ | 2084.51 $\pm$ 674.39 | 983.44 $\pm$ 108.75 | 1.81 $\pm$ 0.56 |
| RPC$_{5bit}$ | 5640.18 $\pm$ 496.67 | 996.15 $\pm$ 60.78 | 1.92 $\pm$ 0.12 |

| HalfCheetah | Rewards | Ep. Length | Entropy Rate |
|---|---|---|---|
| DDPG | **10829.94 $\pm$ 1316.95** | 1000.0 $\pm$ 0.0 | 3.84 $\pm$ 0.10 |
| PASAC$_{k=1}$ | **11134.25 $\pm$ 1398.77** | 1000.00 $\pm$ 0.0 | **-0.02 $\pm$ 0.21** |
| PASAC$_{k=0.5}$ | 11014.73 $\pm$ 816.36 | 1000.00 $\pm$ 0.0 | 0.07 $\pm$ 0.22 |
| RPC$_{2bit}$ | 5105.49 $\pm$ 470.98 | 1000.00 $\pm$ 0.0 | 2.94 $\pm$ 0.12 |
| RPC$_{5bit}$ | 6003.90 $\pm$ 666.42 | 1000.00 $\pm$ 0.0 | **2.55 $\pm$ 0.17** |

| Hopper | Rewards | Ep. Length | Entropy Rate |
|---|---|---|---|
| DDPG | **1080.03 $\pm$ 640.22** | 343.40 $\pm$ 193.73 | 0.89 $\pm$ 0.23 |
| PASAC$_{k=1}$ | 1556.92 $\pm$ 946.95 | 479.17 $\pm$ 300.50 | **-4.28 $\pm$ 0.30** |
| PASAC$_{k=0.5}$ | 2645.24 $\pm$ 1140.11 | 739.70 $\pm$ 332.04 | -3.69 $\pm$ 0.24 |
| RPC$_{2bit}$ | **2667.34 $\pm$ 1091.18** | 732.44 $\pm$ 319.22 | 0.98 $\pm$ 0.15 |
| RPC$_{5bit}$ | 2456.70 $\pm$ 1236.72 | 655.78 $\pm$ 339.75 | 1.19 $\pm$ 0.09 |

---

[17]The Ant-v4 DDPG obtain very high variance rewards and entropies. To get a more conservative baseline, we trained 10 agents and picked the best one in terms of mean reward obtained.

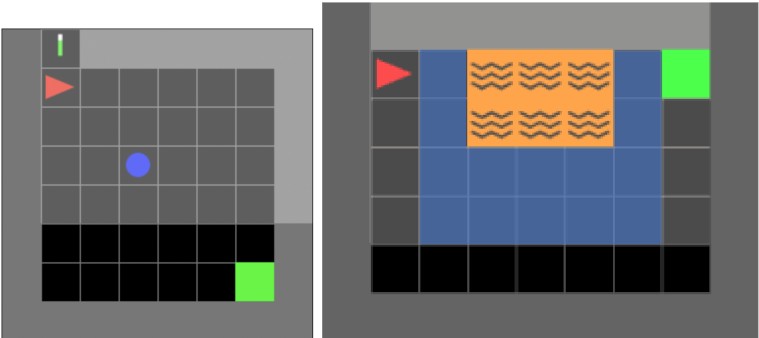

Figure 5: Interactive Tasks. Left is a moving obstacle navigation task where the agent has the option to de-activate the obstacles. Right is a navigation problem where the ground is slippery, resulting in random motion.

## C.2 Interactive Robot Tasks

We created two tasks inspired by real human-robot use-cases, where it is beneficial for agents to avoid high entropy state-space regions. These are based on Minigrid environments Chevalier-Boisvert et al. (2023), see Figure 5. The agents can execute actions $\mathcal{U} = \{\text{forward}, \text{turn-left}, \text{turn-right}, \text{toggle}\}$, and the observation space is $\mathcal{X} = \mathbb{R}^n$ such that the observation includes the robot's position and orientation, the position of obstacles and the state of environment features (*e.g.* the switch). In both tasks, the agent gets a reward of 1 for reaching the goal, or a negative reward of $-1$ for colliding or falling in the lava. Both grid environments were wrapped in a normalizing vectorized wrapper, to normalize observations, but the model data was kept un-normalized.

**Task 1: Turning Off Obstacles**   This task is designed as a dynamic obstacle navigation task, where the motion of the obstacles can be stopped (the obstacles can be switched off) by the agent toggling a switch at a small cost of rewards. The environment is depicted in Figure 5 (left). The switch is to the left of the agent, shown as a green bar (orange if off). The agent gets a reward of $r = 0.95$ if it turns the obstacle off. The intuition behind this task is that agents do not learn to turn off the obstacles, and attempt to navigate the environment. This, however, induces less predictable dynamics since the obstacle keeps adding noise to the observation, and the agent is forced to take high variance trajectories to avoid it. Our Predictability-Aware algorithm converges to policies that consistently disable the stochasticity of the obstacles, navigating the environment freely afterwards, while staying near-optimal.

**Task 2: Slippery Navigation**   The second task is inspired by a cliff navigation environment, where a large portion of the ground is slippery, but a path around it is not. In this problem, the slippery part has uncertain transitions (i.e. a given action does not yield always the same result, because the robot might slip), but the non-slippery path induces deterministic behaviour (the robot follows the direction dictated by the action). The agent needs to navigate to the green square avoiding the lava. If it enters the *slippery* region, it has a probability of $p = 0.35$ of spinning and changing direction randomly. The intuition behind this environment is that PPO agents do not learn to avoid the slippery regions, resulting in higher entropy rates and less predictable behaviours. On the contrary, PAPPO agents consistently avoid the slippery regions. This can be seen in Figure 6.

## C.3 MuJoCo Experiments

We include here the full set of trajectory plots in task space, both for $z$ position of the front tip versus angle of the front tip, and $x$-$y$ velocities of the front tip (See Figures 7, 8, 9, 10, 11 and 12). Additionally, we include in tables 4 and 3 the full numerical results for the simulated agents. Each result is reported computed for 10 independently trained agents, and each agent evaluated over 50 independent trajectories.

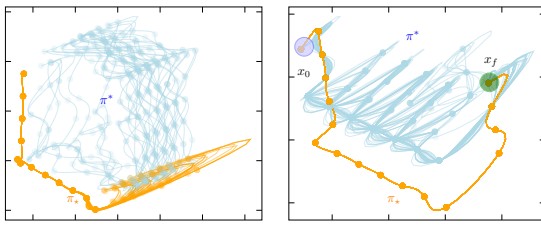

|  (a) Dyn Obstacle Env. | (b) Slippery Nav. Env. |

Figure 6: 2D trajectory projections. Blue is a PPO policy, orange is a PAPPO policy.

Table 3: Results for MuJoCo environments, Stochastic Policies.

| Walker | Rewards | Ep. Length | Entropy Rate |
|---|---|---|---|
| PPO | **2466.27 ± 1329.71** | 638.29 ± 296.82 | 1.19 ± 0.40 |
| PAPPO$_{k=0.05}$ | 2430.12 ± 1199.45 | 626.77 ± 263.34 | 0.36 ± 0.74 |
| PAPPO$_{k=0.1}$ | 2047.30 ± 1106.09 | 551.40 ± 252.01 | 0.06 ± 0.61 |
| PAPPO$_{k=0.25}$ | 1315.47 ± 910.03 | 550.18 ± 296.42 | -1.26 ± 1.79 |
| PAPPO$_{k=0.5}$ | 1072.88 ± 980.41 | 648.38 ± 358.74 | **-1.90 ± 1.16** |

| Ant | Rewards | Ep. Length | Entropy Rate |
|---|---|---|---|
| PPO | 2459.29 ± 1411.75 | 734.79 ± 342.21 | 0.80 ± 0.27 |
| PAPPO$_{k=0.05}$ | **2510.29 ± 1504.11** | 774.54 ± 343.79 | 0.66 ± 0.24 |
| PAPPO$_{k=0.1}$ | 2107.21 ± 1195.23 | 786.88 ± 330.96 | -0.04 ± 1.64 |
| PAPPO$_{k=0.25}$ | 1497.67 ± 854.79 | 950.96 ± 186.39 | -3.54 ± 3.04 |
| PAPPO$_{k=0.5}$ | 1204.47 ± 1076.30 | 987.53 ± 87.62 | **-5.00 ± 1.98** |

| HalfCheetah | Rewards | Ep. Length | Entropy Rate |
|---|---|---|---|
| PPO | 4150.53 ± 1645.16 | 1000.0 ± 0.0 | 1.01 ± 0.34 |
| PAPPO$_{k=0.05}$ | 3575.56 ± 1743.91 | 1000.0 ± 0.0 | 0.46 ± 0.45 |
| PAPPO$_{k=0.1}$ | **4482.48 ± 1679.52** | 1000.0 ± 0.0 | 0.31 ± 0.25 |
| PAPPO$_{k=0.25}$ | 4427.07 ± 1930.60 | 1000.0 ± 0.0 | -0.27 ± 0.74 |
| PAPPO$_{k=0.5}$ | 3962.70 ± 1822.80 | 1000.0 ± 0.0 | **-0.54 ± 0.99** |

**Trajectory Representations** As expected, the observed trajectories for the case of PARL agents present a much less complex (lower entropy) distribution. In particular, for the slippery navigation task where agents have the choice of taking fully deterministic paths, it is even more obvious that the PARL agent chooses to execute the same trajectory over and over, where PPO agents result in a more complex distribution due to the traversing of the stochastic regions.

## C.4 Learning Results

We include the learning curves for the trained agents on all the environments included in the paper.

## C.5 Model Learning

Our proposed predictable RL scheme consists of a model-based architecture where the agent learns simultaneously a model $P_\phi$ for the transition function and a policy $\pi_\theta$ and value functions $V_\xi$, $W_\omega$ for the discounted rewards and entropy rates. Simultaneously to a policy and a value function, we learn a model $P_\phi$ to approximate the transitions (means) in the environment. For this, we train a neural network with inputs $(x, u) \in \mathcal{X} \times \mathcal{U}$ and outputs the mean next state $\bar{y}_{xu}$. The model is trained using the MSE loss for stored data $\mathcal{D} = \{(x, u, y)\}$:

$$\mathcal{L}_y = \frac{1}{2|\mathcal{D}|} \sum_{\mathcal{D}} \left( \bar{y}_{xu} - y \right)^2.$$

We do this by considering $\mathcal{D}$ to be a replay buffer (to reduce bias towards current policy parameters), and at each iteration we perform $K$ mini-batch updates of the model sampling uniformly from the buffer. Additionally, we pre-train the model a set number of steps before beginning to update the agents, by running a fixed number of environment steps with a randomly initialised policy, and training the model on this preliminary data. All models are implemented as feed-forward networks with ReLU activations.

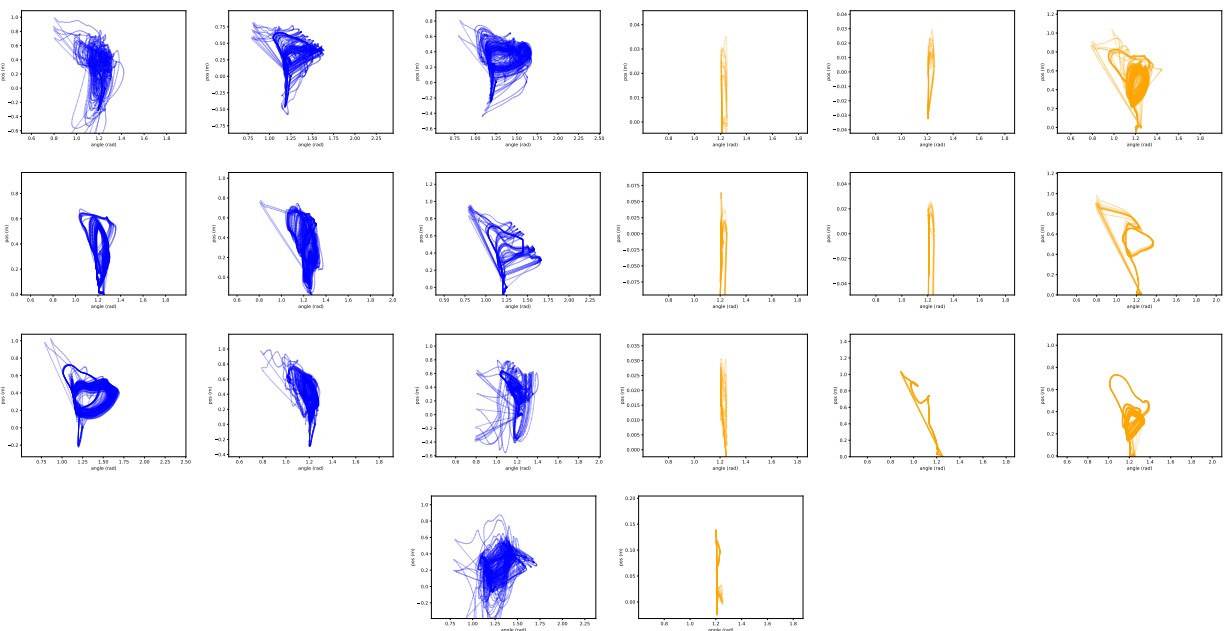

Figure 7: Walker Trajectory Plots, $x$-axis is torso angle in radians, $y$-axis is $z$ coordinate position of the torso. Blue are PPO agents, orange are PARL agents.

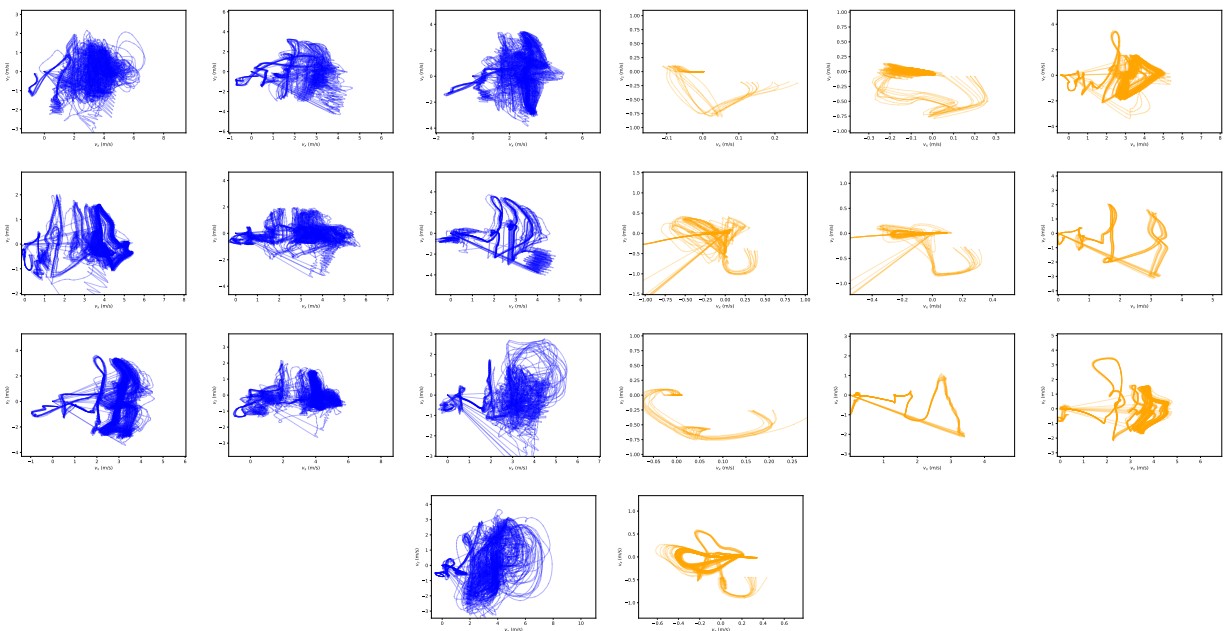

Figure 8: Walker Trajectory Plots, $x$ axis is $x$ velocity, $y$ axis is $y$ velocity. Blue are PPO agents, orange are PARL agents.

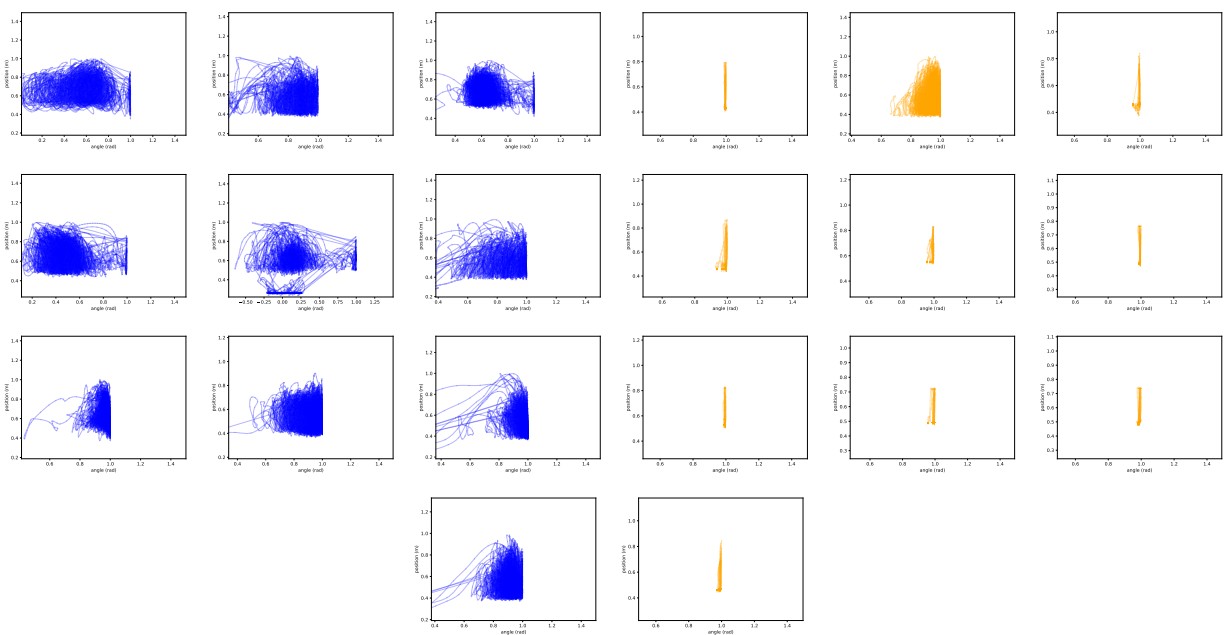

Figure 9: Ant Trajectory Plots, $x$-axis is torso angle in radians, $y$-axis is $z$ coordinate position of the torso. Blue are PPO agents, orange are PARL agents.

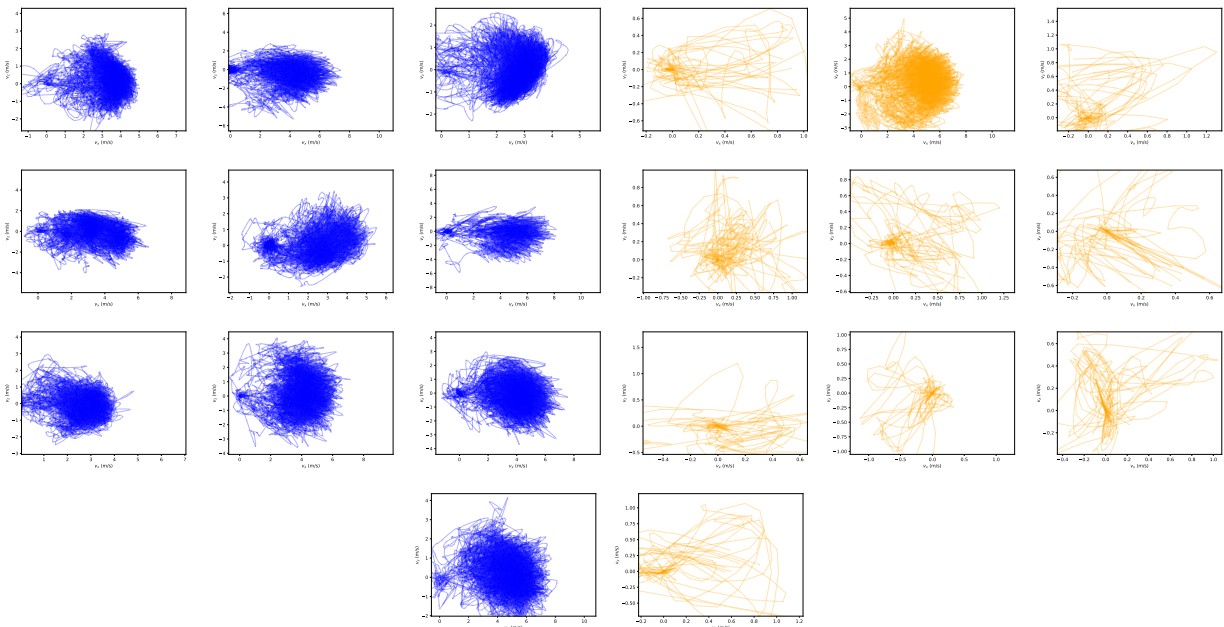

Figure 10: Ant Trajectory Plots, $x$ axis is $x$ velocity, $y$ axis is $y$ velocity. Blue are PPO agents, orange are PARL agents.

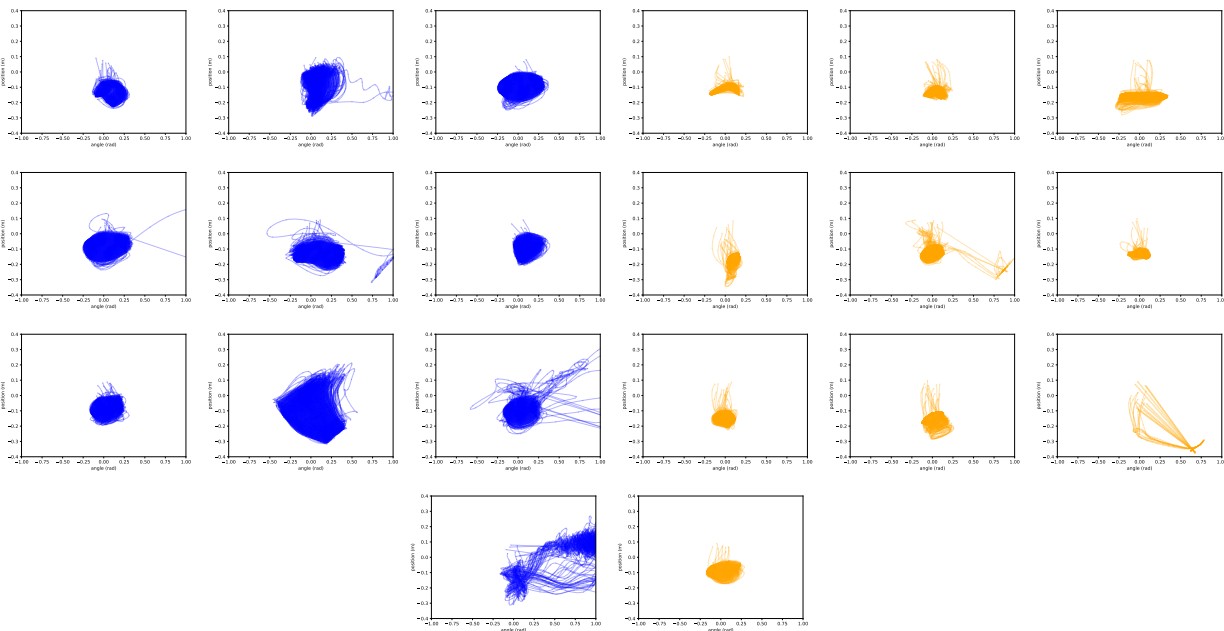

Figure 11: HalfCheetah Trajectory Plots, $x$-axis is torso angle in radians, $y$-axis is $z$ coordinate position of the torso. Blue are PPO agents, orange are PARL agents.

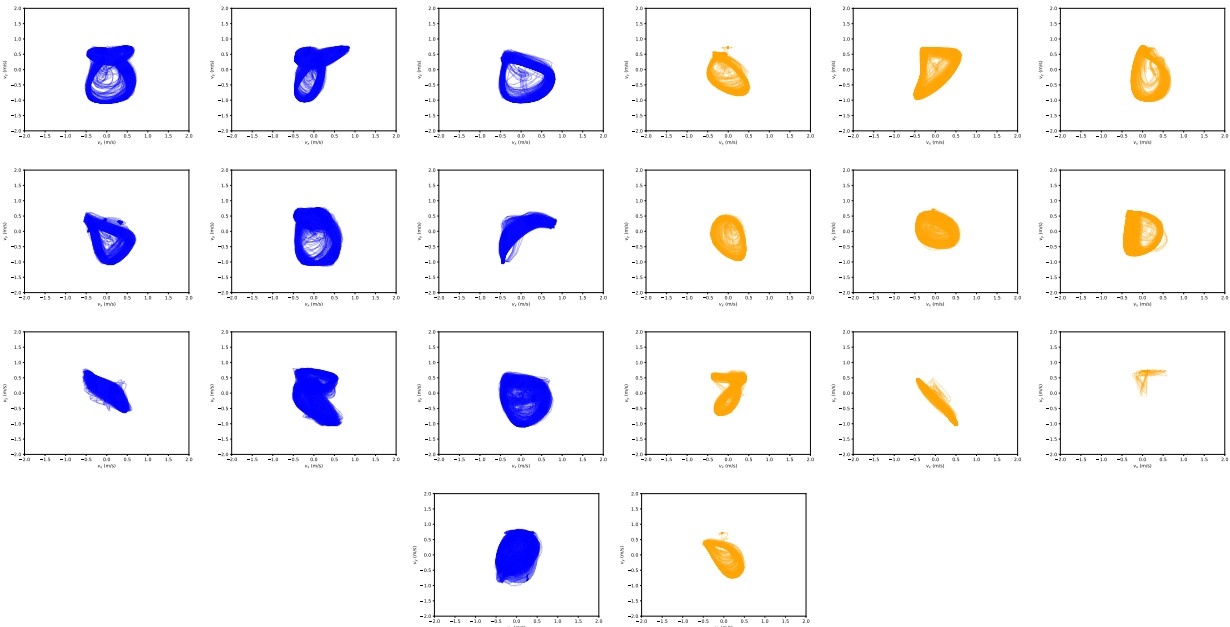

Figure 12: HalfCheetah Trajectory Plots, $x$ axis is $x$ velocity, $y$ axis is $y$ velocity. Blue are PPO agents, orange are PARL agents.

Table 4: Results for MuJoCo environments, Deterministic policies.

| Walker | Rewards | Ep. Length | Entropy Rate |
|---|---|---|---|
| PPO | $3144.66 \pm 1428.29$ | $769.80 \pm 297.99$ | $0.74 \pm 0.43$ |
| $\text{PAPPO}_{k=0.05}$ | $\mathbf{3413.70 \pm 1062.22}$ | $848.21 \pm 218.93$ | $-0.16 \pm 0.79$ |
| $\text{PAPPO}_{k=0.1}$ | $2278.67 \pm 1352.92$ | $598.13 \pm 301.35$ | $-0.20 \pm 0.58$ |
| $\text{PAPPO}_{k=0.25}$ | $1794.17 \pm 1331.01$ | $662.40 \pm 319.09$ | $-1.51 \pm 1.84$ |
| $\text{PAPPO}_{k=0.5}$ | $1212.20 \pm 1201.21$ | $672.42 \pm 368.88$ | $\mathbf{-1.98 \pm 1.12}$ |

| Ant | Rewards | Ep. Length | Entropy Rate |
|---|---|---|---|
| PPO | $3013.54 \pm 1524.71$ | $808.31 \pm 313.55$ | $0.52 \pm 0.38$ |
| $\text{PAPPO}_{k=0.05}$ | $\mathbf{3185.06 \pm 1490.56}$ | $883.68 \pm 260.07$ | $0.27 \pm 0.54$ |
| $\text{PAPPO}_{k=0.1}$ | $2633.50 \pm 1336.18$ | $829.38 \pm 301.95$ | $-0.41 \pm 2.17$ |
| $\text{PAPPO}_{k=0.25}$ | $1611.57 \pm 963.54$ | $964.98 \pm 155.04$ | $-4.77 \pm 3.83$ |
| $\text{PAPPO}_{k=0.5}$ | $1218.27 \pm 1129.12$ | $986.36 \pm 99.72$ | $\mathbf{-6.44 \pm 2.53}$ |

| HalfCheetah | Rewards | Ep. Length | Entropy Rate |
|---|---|---|---|
| PPO | $5192.49 \pm 1181.34$ | $1000.0 \pm 0.0$ | $0.82 \pm 0.31$ |
| $\text{PAPPO}_{k=0.05}$ | $5115.65 \pm 1178.23$ | $1000.0 \pm 0.0$ | $0.27 \pm 0.54$ |
| $\text{PAPPO}_{k=0.1}$ | $5342.42 \pm 1215.20$ | $1000.0 \pm 0.0$ | $0.11 \pm 0.26$ |
| $\text{PAPPO}_{k=0.25}$ | $\mathbf{5686.88 \pm 1096.06}$ | $1000.0 \pm 0.0$ | $-0.37 \pm 0.40$ |
| $\text{PAPPO}_{k=0.5}$ | $4462.63 \pm 1740.05$ | $1000.0 \pm 0.0$ | $\mathbf{-0.97 \pm 0.96}$ |

Figure 13: Training results for Slippery Navigation task.

**Entropy estimation**   We found that it is more numerically stable to use the variance estimations as the surrogate entropy (we do this since the log function is monotonically increasing, and thus maximizing the variance maximizes the logarithm of the variance). This prevented entropy values to explode for environments where some of the transitions are deterministic, thus yielding very large (negative) entropies.

## C.6   Tuning and Hyperparamters

The tuning of PARL, due to its modular structure, can be done through the following steps:

1. Tune (adequate) parameters for vanilla RL algorithm used (e.g. PPO).

2. Without the predictable objectives, tune the model learning parameters using the vanilla hyperparameters.

3. Freezing both agent and model parameters, tune the trade-off parameter $k$ and specific PARL parameters (e.g. entropy value function updates) to desired behaviours.

For our experiments, we took PPO and SAC parameters tuned from Stable-Baselines3 Raffin et al. (2019) and Haarnoja et al. (2018), and used automatic hyperparameter tuning Akiba et al. (2019) for model and predictability parameters. In the implementation we introduced a delay parameter to allow agents to start optimizing the policy for some steps without minimizing the entropy rate. For all hyperparameters used in every environment and implementation details we refer the reader to the project repository.

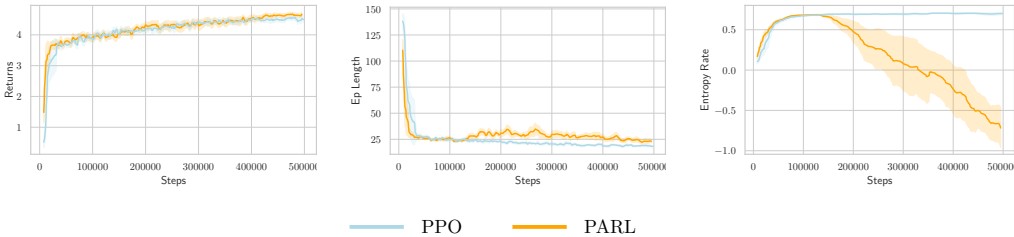

Figure 14: Training results for Obstacle Navigation task.

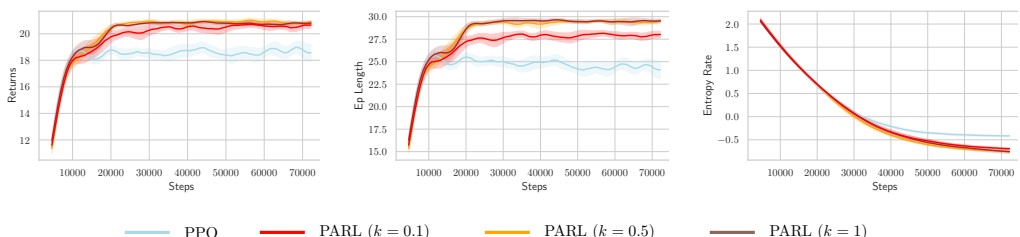

Figure 15: Training results for Highway environment.

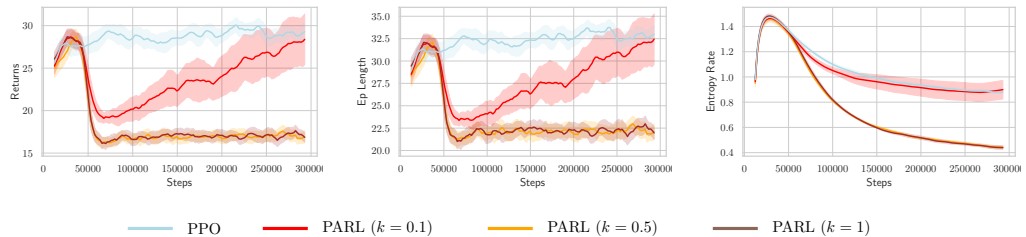

Figure 16: Training results for Roundabout environment.

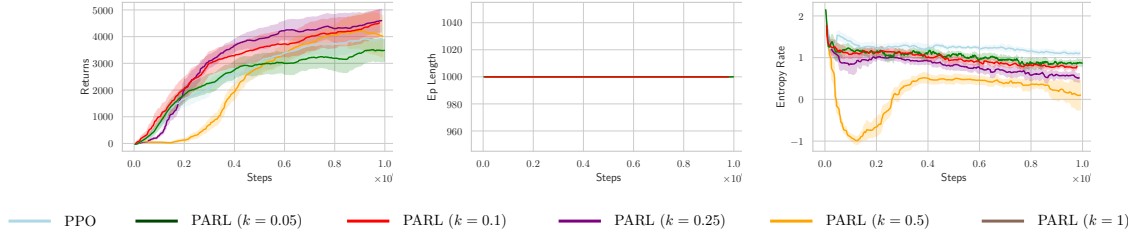

Figure 17: Training results for HalfCheetah-v4 environment.

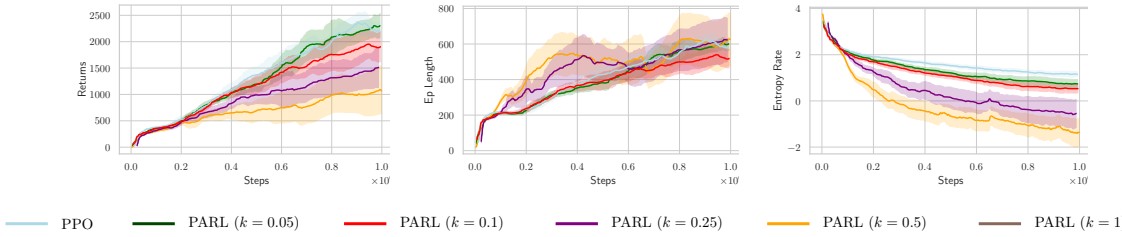

Figure 18: Training results for Walker2d-v4 environment.

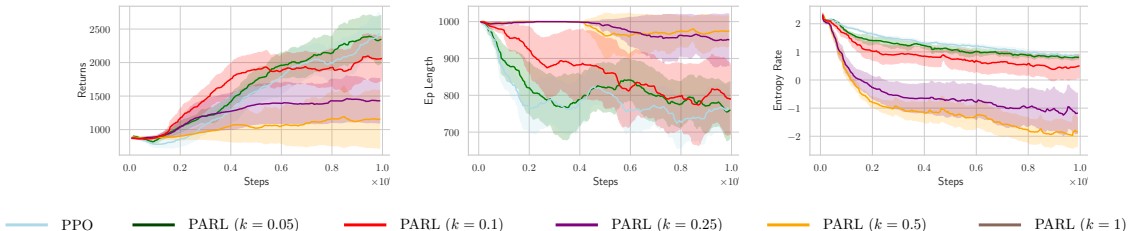

Figure 19: Training results for Ant-v4 environment.

