# OpenReview forum: "Predictable Reinforcement Learning Dynamics through Entropy Rate Minimization"
_TMLR — Accepted by TMLR_

### Review · Reviewer_HXvg · 2025-02-13

**Summary Of Contributions:**

This paper proposed a new RL paradigm to study the complexity of trajectories produced by agents. Simpler trajectories are more predictable and therefore less prone to danger, yet have the potential to achieve the goal same as the complex ones. The authors formulated the predictable RL problem as a trajectory-level entropy regularization problem and presented theoretical proofs justifying the use of surrogate local entropy and approximated dynamics model. Experiments on Mujoco showed that the proposed algorithm led to compact and clustered trajectories.

**Audience:**

Yes

**Claims And Evidence:**

Yes

**Requested Changes:**

Please refer to the weaknesses section for requested changes.

**Strengths And Weaknesses:**

**Strengths**:\
The discussion of predictable RL is novel and interesting. As far as I can recall, there is no work focusing on forcing agents produce simpler and more predictable trajectories. The authors have done a good job on telling the story in the introduction. I especially like how the authors defined the scope: "RL agents are oblivious to the information theoretic loads they generate with their behavior...we do not argue that this is a necessary feature in all agents. We simply argue that it is an interesting feature to consider for general RL agents that can benefit the deployment of RL agents". It is also nice to see that after theoretical proofs, the authors conducted experiments on Mujoco which are not overly simple, showing the efficacy of the proposed method.

**Weaknesses**:\
 - I believe the paper can benefit from moving technical details to the appendix to make the paper less than 12 pages (TMLR short paper). The current draft contains too much technical details that may distract the reader. For example, Proposition 2 is fairly standard where (Borkar, 2009) was used. Its proof can be left to the appendix.
 - Though it is nice to see that the proposed trajectory-entropy minimization led to simpler and clustered trajectories. It is somewhat expected since entropy is a measure of randomness. My concern is that the predictability was not achieved through increasing prediction capability of the agent, but rather through decreasing the randomness of trajectories. This may risk the exploring behavior of the agent in scenarios like multiple reward modes like that in the SAC paper.
 - Another concern is that though the authors critized the entropy-regularization methods that induce higher aleatoric uncertainty, the proposed method falls also into the regularization class. Therefore, the optimal policy is necessarily biased. Does this regularized predictable RL framework "benefit the deployment of RL agents" to problem that requires maximizing task rewards, see ref [1]?

Reference:\
[1] Greedy actor critic, ICLR 2023

---

> ### Author Response · Authors · 2025-03-03
> **Reply to Reviewer's Points**
>
> (Please note the updated version of the paper will be uploaded after all reviews have been received)
> We would like to thank the reviewer for the time spent in the review, and the constructive comments. Let us address each of the questions and weaknesses, with the corresponding changes to the paper:
>
> **1. Moving proofs to appendix:**
> We agree with the reviewer. To make the paper more readable (and fit under the TMLR short paper length) we have moved some of the more convoluted and less relevant proofs to the appendix.
>
> **2. Predictability Discussion:**
> We would like to clarify first that by “more predictable” we mean “easier to predict by standard inference algorithms”, or in other words, a prediction model trying to infer the next position of the agent given the past history of positions would consistently make lower prediction errors. The established notion for predictability in information theory is entropy, such that a lower-entropy stochastic process is easier to predict by standard inference algorithms. By enforcing lower entropy in agents’ trajectories, this makes them simultaneously less random (again, in terms of entropy) and therefore easier to be predicted with lower errors. Thus, decreasing randomness (in terms of entropy) and increasing prediction capability are two sides of the same coin.
> In terms of exploration, this did not seem to be a problem in the different environments tested, for the following reasons. First, the agent needs to rely on having an accurate model to be able to effectively reduce trajectory entropy (as this information is propagated into the surrogate entropy value function); in fact, through proper tuning, we can ensure that the models are learned slower than the reward value function, for example. This alone pushes the agents to reduce their trajectory entropy after the problem has been explored (since exploration is a requirement for having accurate models). In any case, if this were to cause exploration problems, there are heuristic solutions, such as slowly activating the entropy value function coefficient $k$ in the policy gradient only after a certain amount of steps, or casting the two-objective optimization as a lexicographic scheme where we guarantee that the rewards are sufficiently maximised before the trajectory entropy is minimised.
>
> **3. Regularisation and bias:**
> Indeed let us clarify this point. Our statement did not attempt to criticise all regularisation schemes; these are useful and in fact necessary in many optimization problems, as well as in reinforcement learning. Indeed, our proposed method results in a combined optimization objective which can be interpreted as a maximisation of rewards with a (negative) trajectory entropy regularisation. For problems where ensuring convergence to a reward-maximising optimal policy is critical, any regularisation would be undesirable. However, one can argue that in many robotic applications (e.g. self driving vehicles) where rewards are specifically crafted to induce solving some task, there may be several ‘acceptable’ (sub) optimal policies, and from those our method would allow to converge towards the policy that induces lower randomness in the observed behavior of the agent.

---

### Review · Reviewer_myg1 · 2025-02-17

**Summary Of Contributions:**

The paper proposes a method to enforce predictability of RL behaviours, defined as polices inducing low-entropy distributions over trajectories. In order to do so, the authors propose a (negative) reward shaping term based on two fundamental steps: first,  a reformulation of the entropy over trajectory distributions as an average reward problem, then an objective surrogate that upper-bounds the true one in order to guarantee non-degradation over the true objective. The derivation is accompanied with a series of theoretical result to guarantee that the proposed method is well-founded and a series experiments confirming that the proposed method does induce trajectories with low entropy rate.

**Audience:**

Yes

**Broader Impact Concerns:**

No concerns

**Claims And Evidence:**

Yes

**Requested Changes:**

Major Concerns and Changes:
- The main theoretical contributions are often not presented in a clear and self-explanatory way. Every step and/or result needed for understanding the derivations should be reported.Too often, results from other works are cited but not reported, making the overall understanding hard. For instance,
  - Lemma 1: Theorem 3 is left undefined in the main paper, even thought it seems rather central;
  - Proposition 1, "Thus, from the Fannes–Audenaert inequality Fannes (1973), for the special case of diagonal density matrices (representing conventional probability distributions), we obtain:" is obscure to me, also because the inequality is usually applied to Von Neumann entropies, as far as I know;
  - Proposition 2: I am not sure how a bound on infinity norm could turn into an equality with no dependency on x as the first passage. Also, "Therefore, by Theorem 6 in (Borkar, 2009), the iterates converge to some point" is a rather obscure sentence to the average reader which might need way more discussion.
- I am not sure to have understood how the all discussion of Section 7 is relevant at that point of the discussion.
- Experiments:
  - Deterministic policies in PPO are used as comparison, but algorithms that learn deterministic policies directly (DDPG) were not. I would expect to see them as a comparison.
  - How were entropy rates computed in practice should be discussed.
  - In the autonomous driving task, a negative reward is usually given in case of collision and lower rewards would actually result in unsafe behaviours, how this is related to the whole narrative should be discussed.



Some Minor Concerns and Changes:

- (Page 1) Explain why "Higher entropy rate implies more complex and less predictable trajectories, and vice-versa." as it is not obvious to all the readers.
- (P2) In figure 1 the needed policy seems to be deterministic (rather than the distribution over trajectories), if this is the case I would comment on why.
- (P3) Delta notation for distributions not introduced.
- (P4) typo--> 0:T-1 should be 0:t-1 in the joint entropy, I would also merge section 3 with the next one
- (P4) I believe the definition of conditional entropy can be made more rigorous by defining what p stands for and where the dependency on the policy in maintained.
- (P5) the derivation of Eq 4 can be made more structured and formal, being of central interest. For instance, how P_\pi relates to the p of the previous definition, how the dependency on the joint probability over trajectories of the conditional entropy is implied to the Markov kernel in Eq 3.
- (P8) typo --> I think S^\pi_\phi should be dependent on P_\phi. And the Cited Corollary 1 is actually Proposition 1.
- (P10) Remark 3 is hard to read as it is.

**Strengths And Weaknesses:**

Strength: The use of entropic functional to enforce predictable behaviors is rather new, additionally the derivations and results are rather on standard (to me).

Weakness: while being a positive as well, the rather non-standard derivations are rather hard to follow, often based on some results not properly introduced.

Some high-level concerns about the main motivation and context:
- It is still just at an intuitive level to me how a low-entropic distribution over trajectories should be inherently more predictable. I think a broader discussion of what to do with almost deterministic trajectories (once reached) should be added.
- The authors argue that "[randomized actions] often makes it challenging for other agents and humans to predict an agent’s behavior, triggering unsafe scenarios" and suggest to enforce deterministic distributions over trajectories. This is clearly different from enforcing deterministic policies only, but I think justifying why a simple DDPG architecture (for instance) would not suffice is essential. Also, testing against less deterministic environments would be needed, as in (almost) deterministic environments deterministic policies induce low-entropy distributions over trajectories.

---

> ### Author Response · Authors · 2025-03-03
> **Reply to Strengths and Weaknesses points**
>
> (Please note the updated version of the paper will be uploaded after all reviews have been received)
> We would like to thank the reviewer for the very detailed and useful review.
> ## Strengths and Weaknesses
> **On lower entropy distributions being more predictable:** We would like to clarify first that by more predictable we mean ‘easier to predict by standard inference algorithms’, or in other words, a prediction model trying to infer the next state of the agent given the past history of trajectories would consistently make lower prediction errors. Now, the established notion for predictability in information theory is entropy, such that a lower-entropy stochastic process is easier to predict by standard inference algorithms. For example, assuming the trajectory distribution would be a normal distribution, lower entropy is equivalent to lower variance, and it is clear that inference algorithms would be able to predict more accurately a lower variance distribution. The same applies to other distribution families, including discrete distributions. In the limit, minimum entropy implies that the agent follows the same deterministic trajectory over and over, which can be predicted perfectly with past data (since it is deterministic). We will add this detailed discussion to the paper discussion section.
>
> **On why deterministic policies are not sufficient:** As the reviewer points out, a deterministic policy does not guarantee deterministic trajectories. Recall that, most often, trajectories are sequences composed not only by the agent’s internal variables, but also by environment variables. As such, PARL pushes agents to avoid high-entropy sub-spaces as well, i.e. spaces where the environment might induce lots of randomness (see e.g. the grid world results in Appendix C2, where agents learn to turn off environment randomness through PARL). On the contrary, a standard deterministic policy that does not take entropy into account, might lead to subspaces with high uncertainty. As an example, in the autonomous driving experiments, DQN learns deterministic policies by default, and yet does not guarantee low variance on agents’ trajectories. In this example, a deterministic policy, by optimising rewards, attempts to drive fast through traffic, generating much more diverse (and random) trajectories. PARL, on the other hand, sacrifices some reward (e.g. driving a bit slower) in exchange for navigating into a more predictable environment. We have included animations of the highway environment as an illustration of this behavior.

---

> ### Author Response · Authors · 2025-03-03
> **Reply to Requested Changes**
>
> ## Proposed changes
> ### Clarification of Theoretical Derivations:
> - We agree with the reviewer that some derivations need more clarification. We have now expanded these sections. In particular, we have provided an introduction to Theorem 3 in the main text and referred to the appendix for details, and clarified how the Fannes–Audenaert inequality applies in our setting (in particular, the main point is to interpret the density matrices as diagonal matrices with probabilities $P$ and $P_{\phi}$; then the Von Neumann entropy coincides with the Shannon entropy). On Proposition 2, indeed it suffices that the infinity-norm bound depends solely on \epsilon, for the results from Borkar to apply (i.e., the equality $S^\pi(x,u)=S^\pi_\phi(x,u) + \eta(\epsilon)$ is indeed false, but also not needed for the proof; the only thing that is needed is the bound on the infinity norm). We have corrected this in the paper, and have added more detail to the application of the results from Borkar.
> ### On section 7:
> The main point of Section 7 is to show that our method naturally extends to the case of average reward objectives, through a modified single value function. We are happy to clarify this better in the paper.
>
> ### Experiments:
> - We did include deterministic algorithms in the case of the autonomous driving experiments (DQN). We have now also run DDPG benchmarks for the mujoco experiments (these are included in the off-policy results table in the appendix, since it is an off-policy algorithm). Observe that (as discussed in above points) deterministic policies do not necessarily yield lower entropy rates. In some cases these are even higher than PPO benchmarks, which can be explained due to the fact that on-policy algorithms seem to be easier to predict (numerically, by deep learning prediction models), likely since the prediction models are being trained with on-policy data.
> - We have added a short subsection explaining how we compute the entropy rates given the prediction models learned (in section 5.2).
> - In the autonomous driving task, even though PARL might lead to lower rewards, it appears that it leads to less (or no) collisions, as it enforces lower entropy rates, which seems to coincide with more predictable and safer driving patterns. Besides, notice that, in this example, lower reward does not necessarily mean more collisions, but it might mean that the agent deviates from the desired speed (which is indeed what happens with PARL). Nonetheless, generally speaking, in environments where rewards encode safety constraints, then adding any secondary objective (such as entropy regularization) to the policy optimisation might indeed introduce safety risks; a way to address this would be through a lexicographic approach to ensure that minimising entropy rates does not cause an unsafe drop in rewards.
>
> ## Minor Concerns:
> - We have expanded the discussion on the first page on how predictability relates to entropy (including some of the points raised above). Figure 1 simply illustrates that PARL pushes the policy synthesis towards a policy that induces a ‘narrower’ distribution of trajectories (which could be a deterministic trajectory depending on the environment), but this is not dependent on the policy itself being deterministic. We have corrected the typos pointed out. We have properly introduced $p$ in page 4, and expanded the derivation of eq (4) given the suggestions.  We have also simplified remark 3.

---

### Review · Reviewer_r3hW · 2025-04-13

**Summary Of Contributions:**

The paper proposes using an information-theoretic objective to minimize the entropy of the state visitation distribution in RL agents to make their behaviour more predictable, evaluating the approach on mujoco and driving environments.

**Audience:**

Yes

**Claims And Evidence:**

No

**Requested Changes:**

Please address my concern regarding the accuracy of referring to this approach as maximizing the predictability of the *agent's* behaviour.

I think the paper's empirical contribution would be significantly stronger if the authors were able to demonstrate an environment where the method unambiguously helps to improve performance, and where this improvement can be well-explained by the nature of the source of transition-entropy in the environment (for example, noisy states being harder to learn an optimal policy on or less controllable).

**Strengths And Weaknesses:**

**Strengths**

The paper proposes an interesting idea which runs counter to prior work on entropy maximization in RL.

The approach overcomes some technical difficulty in formulating an entropy-minimizing objective.

The algorithm is clearly presented.

The paper evaluates on both deterministic and stochastic variants of environments, and considers multiple environments from a well-established benchmark (Mujoco).

Under a number of assumptions which, while not guaranteed to hold in practice, are a reasonable starting point for analysis, the method is shown to converge.

The visualization of the differing trajectories between the two methods provided a nice qualitative example of the behaviour induced by the algorithm. Quantitatively, the results on the entropy under different coefficients suggest that the objective is being successfully optimized.


**Weaknesses**

The claim that the proposed method increases the predictability of the agent’s behaviour is inaccurate. As a trivial example, an agent in an environment where turning left on the first timestep takes the agent into a region where the transition dynamics become deterministic and turning right causes the agent to enter a region where the dynamics are noisy can maximize the PAPPO objective by always turning left, even if it then follows a uniform random policy in every other state.

It seems to me that the proposed algorithm is optimizing for predictability of the environment, encouraging the agent to visit states where the environment dynamics are closer to deterministic. This type of objective seems like it would be especially useful in partially observable environments, which to the best of my knowledge are not studied in this paper. I would be surprised if the mujoco tasks evaluated were really the best domain to showcase the benefits of entropy minimization. The multi-agent driving environments get closer to this property (indeed, I could imagine that this type of entropy penalty might induce more “defensive” driving compared to the baseline agent), but not consistently, as evidenced by the performance degradation in roundabout.

The method generally appears to at best provide a very modest improvement in performance (which is still within the confidence interval of the baseline PPO agent) and at worst significantly degrade performance for larger entropy penalties.

The method requires a world model, and I would guess that its success depends heavily on properties of this model. In particular, in rich-observation settings I would expect that the generalization of the learned model would heavily influence the agent’s exploratory behaviour. I did not see any additional investigation into the role of the world model choice on performance, which seems important to understanding the method.

---

> ### Author Response · Authors · 2025-04-15
> **Reply to review**
>
> We would like to thank the reviewer for the positive feedback and the useful comments and questions. Let us address each of the concerns and suggested modifications one by one.
> ## Weaknesses
> - *Clarification on Predictability:* We agree that the term ‘agent behaviour’ may be ambiguous. While the definition of the entropy rate of an MDP (Section 3) does account for policy entropy (via the induced Markov Chain entropy, a deterministic policy would induce a deterministic chain with zero entropy, and a random policy would induce a stochastic chain, with non-zero entropy), it is true that the lower bound derived in further sections de-couples this, and only considers the state transition entropy. While we show that this is a formal bound for the true entropy rate, as well as the cases where the bound is tight (for e.g. deterministic policies), we agree with the reviewer that the surrogate entropy could be low if the agent is in a fully deterministic region of the state-space and still uses randomized policies. We would like to argue, however, that this feature is not a problem for several reasons.
>     1. This allows us to decouple objectives and allows PARL to not directly discourage exploration (which, in PPO is induced by regularizing policy entropy). An agent could simultaneously have high policy entropy allowing it to explore, and still be learning what regions of the state-space have less uncertainty (and gravitate towards them). We conjecture that this is one of the main reasons why exploration was not an issue when applying PARL. As a different interpretation, this decouples the impact of different sources of randomness in the agent learning process; if there is a decrease in entropy rate, there is no ambiguity on whether it comes from policy or environment randomness.
>     2. In fully observable settings, optimal policies are guaranteed to be deterministic. Furthermore, some RL algorithms directly produce deterministic policies (DDPG, DQN…). The randomness introduced in the agent behaviour by the policy is in these cases decaying (or completely removed) at policy inference time, and behaviour predictability and environment randomness become completely coupled.
>
>     To reduce ambiguity we have replaced ‘behavior’ by ‘trajectories’ in the abstract, and have added clarification on the text where we first mention ‘predictable agent behaviour’ (footnote in page 1), to specify that we refer to randomness in the sequence of states the agent traverses/observes. Furthermore, we have also added a comment below Lemma 1 to clarify whether the policy randomness affects the entropy rates.
> - *POMDPs:* While we agree that PARL's benefits might be particularly useful in POMDPs, considering partially observable environments introduces a whole new set of complexities. The entropy rate of a POMDP (if measured on the state sequence, not on observations) is influenced by the policy given an observation, observation probability given a state, and the state transition, resulting in a much more difficult object to estimate (see e.g. Savas et al 2021). We believe the theoretical and practical considerations of minimising entropy rates in POMDPs (in model-free settings) would require extensive follow-up work (and will be a very interesting future direction).
> - *MuJoCo Experiments:* We chose to implement Mujoco experiments as it represents a standard, well-established benchmark suite for continuous control RL. Our goal was to demonstrate the method's applicability and effect on trajectory distributions within these commonly used environments, allowing for comparison against strong baselines like PPO, and evaluate whether PARL causes performance degradation. The visualizations (noted as a strength) help illustrate the effect even in these domains. For demonstration purposes, please note that the self-driving examples can be considered partially observable as the agent only observes the vehicles in its field of view, and these appear/disappear as the agent moves.
> - *Influence of the model:* While we agree that, empirically, prediction models will have an impact on the effectiveness of the method, we implicitly rely on the assumption that the models are able to (up to some reasonable accuracy) predict the dynamics. We agree that for very complex RL environments a careful study of the model choice would be beneficial, but we see this as complementary and building on our work.

---

> ### Author Response · Authors · 2025-04-15
> **Reply to Requested Changes**
>
> ## Requested changes
> - See above for the change regarding the term agent behavior.
> - *Objective and Trade-off:* Our primary goal with PARL is not necessarily to improve task reward over a well-tuned baseline (like e.g. PPO), but rather to introduce a mechanism for controlling the predictability (trajectory entropy) of the agent's behaviour. As noted by the reviewer, our quantitative results show that the entropy objective is being successfully optimized, but this introduces a bias (as discussed further with Reviewer HXvg below) towards lower-entropy trajectories, which inherently involves a trade-off with pure reward maximization. For applications where predictability, consistency, or ease of monitoring are important (e.g. safety-sensitive deployments), lower trajectory entropy, even without exceeding baseline reward performance, can be a valuable feature. Finally, the effect mentioned in the last paragraph of the review is precisely highlighted in the mujoco experiments, where we observe (modest) increases in rewards obtained for PARL agents, indicating that having more predictable trajectories actually helps control the system. Note, however, that this is empirical and problem-dependent, and we cannot claim that PARL helps improve reward maximising performance in general RL algorithms.

---

### Decision · Action_Editor_UiK1 · 2025-05-27

**Recommendation:** Accept as is

**Comment:**

The paper deals with a problem formulation, regularized RL with the negative entropy rate, that is, to the best of my knowledge, original. I believe this formulation is worth studying and the contribution has potential, a sentiment that is shared by the reviewers. The paper is solid and does not present major flaws, although some area of improvements have been extensively reported in the reviews.

I am proposing acceptance of the paper as is, although I am advising the authors to read carefully the reviews when drafting their final version. Especially,
- to take seriously the comment from Reviewer r3hW and adjust the narrative as "trajectory predictability" rather than "action predictability";
- expand the discussion on why this formulation may be particularly valuable in POMDPs (as commented in response to Reviewer r3hW).
Most of the concerns of other reviewers seem to have been incorporated in the manuscript already.

Finally, the authors may want to discuss the relation between their work and "SMiRL: Surprise Minimizing Reinforcement Learning in Unstable Environments" (Berseth et al., 2021), which is studying a formulation with some similarities (and important differences, mainly the entropy of the state distribution instead of trajectories and surprise minimization as an unsupervised objective instead of regularization).

**Audience:**

The paper can draw the interest of the RL community.

**Claims And Evidence:**

The submission studies RL with a regularization related to the negative entropy rate (of the trajectories induced by the agent's policy). The work claims it is possible, under some circumstances, to increase predictability of the RL policy while keeping near optimality, which is validated numerical in Mujoco tasks.

---

> ### Author Response · Authors · 2025-06-02
> **Reply to decision**
>
> Dear Editor,
>
> We would like to thank you for the time and effort spent in the review process. We have updated the camera ready version with the last suggested points.
>
> Kind regards,
>
> The authors